# How can we trust TROPOMI based Methane Emissions Estimation: Calculating Emissions over Unidentified Source Regions

Bo Zheng[1], Jason Blake Cohen[1,2*], Lingxiao Lu[1], Wei Hu[1], Pravash Tiwari[1], Simone Lolli[3], Andrea Garzelli[4], Hui Su[5], Kai Qin[1,2*]

[1]School of Environment and Spatial Informatics, China University of Mining and Technology, Xuzhou 221116, China

[2]Shanxi Key Laboratory of Environmental Remote Sensing Applications, China University of Mining and Technology, Xuzhou 221116, China

[3]CNR-Institute of Methodologies for Environmental Analysis (IMAA), Contrada S. Loja, Tito Scalo, 85050, Italy

[4]Department of Information Engineering and Mathematics, University of Siena, 53100 Siena, Italy

[5]Department of Civil and Environmental Engineering, Space Science and Technology Institute, The Hong Kong University of Science and Technology, Clear Water Bay, Hong Kong SAR

*Correspondence to*: Jason B. Cohen and Kai Qin (jasonbc@alum.mit.edu ; jasonbc@cumt.edu.cn ; qinkai@cumt.edu.cn)

**Abstract**. We propose a novel method for computing the effects of TROPOMI observational uncertainties on emissions calculation arising from the nonlinearity of the gradient terms and non-biased filtering in space and time. Application using TROPOMI $XCH_4$ data in clean areas of Western China with long-term WMO background observations quantifies a minimum detectable emission threshold of 0.3 μg/m²/s, lower than existing community thresholds using TROPOMI. By combining threshold-based and stochastic approaches that incorporates pixel-by-pixel and day-by-day $XCH_4$ uncertainties, we identify and filter physically unrealistic emission values in both space and time. The resulting emissions reveal both missing sources and a 5% emission bias caused by the nonlinearity of the gradient term. Validation was performed by applying the method to the Permian Basin, where comparisons with airborne observations demonstrate the method's ability to align with independent datasets. The importance and implications of our results are related to this being a new methodology for methane emissions estimate from TROPOMI which enables precise identification of emission sources and improved handling of observational noise, offering a more accurate framework for methane emission monitoring across diverse regions using existing satellite platforms. Our results yield a non-negative emissions dataset using an objective selection and filtration method, which includes a lower minimum emissions threshold on all grids and reduction of false positives. Finally, the new approach can be adopted to other satellite platforms to provide a more robust and reliable quantification of emissions under data uncertainty that moves beyond traditional plume identification and background subtraction.

## 1 Introduction

Methane (CH₄) is a potent greenhouse gas with a global warming potential (GWP) estimated to be 28-34 times that of carbon dioxide over a 100-year period and 84-87 times on a 20-year time scale (Hu et al., 2024; Vanselow et al., 2024; Liang et al., 2023). Major sources of methane include fossil fuel extraction, agricultural activities (such as rice paddies and livestock), waste disposal, and wetlands (Vanselow et al., 2024). Since the start of the Industrial Revolution, methane concentrations have been observed to significantly increase (Liang et al., 2023; Erland et al., 2022), with periods of both larger growth and slower or no growth contained within, contributing significantly to the present amount of observed global warming (Vanselow et al., 2024; Erland et al., 2022; Prinn et al., 2001). Due to its relatively shorter atmospheric lifetime compared with many other greenhouse gases, accurately quantifying and controlling methane emissions is crucial in addressing global warming in the near-term (Erland et al., 2022; Liu, J et al., 2024a).

In recent years, with the development of satellite remote sensing technology, various platforms from space have become more relevant at aiding the monitoring of methane concentrations in the atmosphere (Gao et al., 2023; Nesser et al.,2024). Among them, TROPOspheric Monitoring Instrument (TROPOMI), Greenhouse Gases Observing Satellite (GOSAT) and SCanning Imaging Absorption spectroMeter for Atmospheric CHartoghraphY (SCIAMACHY) have been used for its retrieval of a physical atmospheric CH₄ column retrieval (XCH₄), allowing indirect validation by surface networks (i.e., Advanced Global Atmospheric Gases Experiment (AGAGE) and World meteorological Organization (WMO)) as well as upward looking surface observational networks (i.e., Total Carbon Column Observing Network (TCOON)) (Hu et al., 2016; Parker et al.,2011; Frankenberg et al., 2005). TROPOMI in particular offers a combination of daily XCH₄ retrieval, global coverage, and high spatial resolution (Erland et al., 2022; Liu, M et al., 2024; Gao et al., 2023; Nesser et al., 2024; Hu et al., 2016). Due to its extensive use to monitor other atmospheric constituents impacting air pollution, there is extensive work to estimate the uncertainty of TROPOMI products including: $NO_2$ from 10% to 40% (Boersma et al., 2018; Pollard et al., 2022), black carbon aerosol (BC) from 20%-50% (Vignati et al., 2010; Tiwari et al.,2023,2025), and CO starting from a minimum of 10% to 20% and upwards, without a well-defined upper range (Sha et al., 2021). Additionally, XCH₄ from this platform has a significant number of pixels with missing data (Schneising et al., 2023; Qu et al., 2021; Hachmeister et al., 2022; Frankenberg et al., 2016; Hu et al., 2018). Therefore, XCH₄ is expected to also have a significant uncertainty, due to at least the following factors: sensor and waveband resolution issues (Hu et al., 2018), atmospheric conditions which reduce the already relatively weak incoming shortwave infrared radiation such as clouds and aerosols (Balasus et al., 2023; Liu, Z et al., 2024; Liu, J et al., 2024b), and surface reflectivity (Balasus et al., 2023).

Computing emissions from concentration requires an additional step that includes knowledge of the wind, atmospheric transport and diffusion, in-situ processing and other processes (Cohen and Prinn 2011; Cohen et al., 2011). There are many studies which have used various complex approaches such as chemical transport models (Hancock et al., 2025; Nesser et al., 2024; Varon et al., 2023) and simple approaches such as the Gaussian integral method (Schneising et al., 2020), Divergence method (Liu, M et al., 2021; Veefkind et al., 2023), mass balance method (Hu et al., 2024) to attempt to estimate methane

emissions from TROPOMI in near real time. However, due to the uncertainties in the retrieved TROPOMI data and wind
fields, the computation of gradient term(s) necessary to compute emissions may lead to significant non-linearity (Cohen and

Prinn 2011; Cohen et al., 2011; He et al., 2024), thereby generating negative emissions and smaller emission values. Hancock
et al. (2025) prevented non-physical negative emissions by inverting prior emissions using the lognormal error probability
density function (PDFS), Schneising et al. (2020) defined the background from a $2° \times 4°$ headwind box and determined negative
emissions or very low emissions for many days in the Permian basin, despite strict filtering criteria for background observations.
Veefkind et al. (2023) employed background subtraction to filter out negative emissions and smaller emission signals in

Permian basin and believed that the intense variations in topography and surface albedo may lead to significant negative
emissions. However, these studies did not further analyze the causes of negative emissions. Simple background filtering
methods cannot effectively remove all emission noise, and may in fact leave noise contributing to overestimation of actual
emissions. Therefore, finding ways to address how observed and modeled uncertainties lead to the robustness of inverted
methane emissions is a necessary and essential step to gain trust in resulting emission quantification and usefulness for

attribution (Li et al., 2025; Tiwari et al., 2025). To gain confidence, such emissions should include methods which are both
unbiased and robust, and are capable of identifying sources both known and unknown, monitoring emissions from those
sources, and provide validation of which sources are actually being reduced or eliminated (Hemati et al., 2024).

In this study, we quantify $CH_4$ emissions in a clean area with a long-term WMO observational station (Waliguan) using
a flexible mass-conserving method that explicitly accounts for the impacts of $XCH_4$ observational uncertainties on the gradient

terms and their influence on emission inversion. Selecting a region devoid of large emission sources is critical, as it allows for
an objective demonstration of the effects of white noise generated by the nonlinearity of the gradient term on emission
calculations. To address this, we introduce unbiased thresholds and filters to systematically separate genuine emission signals
from white noise, effectively eliminating physically implausible emission values in an unbiased manner. Our results
demonstrate that all identified emissions correspond to spatial and temporal locations where emissions are expected to occur,

with no computed emissions detected in regions where emissions should not exist. To further validate the robustness and
applicability of our method, we applied it to the Permian Basin, the largest and fastest-growing oil and gas-producing basin in
the United States, located in western Texas and southeastern New Mexico. By aligning our results with known emission source
locations in the region, we successfully captured the majority of emission signals, including many smaller emissions that were
previously undetected using TROPOMI, thanks to our method's lower detection threshold. Comparisons with conventional

approaches, which simply remove negative concentration or emission values, reveal that our method provides a better match
with observational data, demonstrating improved stability, accuracy, and a reduction in systematic bias. This highlights that
this approach is capable of delivering reliable and precise emission estimates, even in areas with complex sources.

## 2 Data and Methods

### 2.1 study data

TROPOMI inverts daily measurements of column-averaged dry-air atmospheric CH$_4$ column mixing ratio based on a retrieval around the NIR and SWIR absorption bands, with overpasses occurring daily around 13:30 local time. The XCH$_4$ retrieval used herein (version 2.4.0 Level-2) relies on a physical algorithm that factors in surface and atmospheric scattering. In this study, we use the methane total column-averaged dry-air mole fraction XCH$_4$ of TROPOMI from May 2018 to December 2023 in Waliguan (35.5° to 37.1°N latitude, 100.1° to 101.7°E longitude) (Hu et al., 2018; Lorente et al., 2021;

Landgraf et al., 2024) and from 2019 to 2020 in the Permian Basin (31° to 33°N latitude, 101.3° to 104.3°W longitude). To ensure high quality data, grids with quality assurance less than 0.5 were removed. Swath-by-swath data were resampled day-by-day onto a standard latitude/longitude grid of 0.05° x 0.05° using HARP (http://stcorp.github.io/harp/doc/html/index.html).

    Wind speed and direction were obtained from the European Centre for Medium-Range Weather Forecasts ERA-5 reanalysis product. Due to the elevation and overpass time, 550-750 hPa level meteorology products were used in the Waliguan

area and 750-950 hPa range products in the Permian Basin, both using an average value based on of local 13:00 and 14:00 UTC (Hersbach et al., 2023).

    Daily ground observations of CH$_4$ made for more than three decades at Waliguan are obtained from World Data Centre for Greenhouse Gases. The source coordinates and emission rates of methane in the Permian Basin were obtained from an aircraft observational study by Cusworth et al. (2021).

High frequency CH$_4$ flux was measured nearby a gas extraction vent of a high gas coal mine in Changzhi, Shanxi Province using the eddy correlation method, using CSAT-3 anemometer and LI-7700 and Universal open Path gas analyzer at 10 Hz as published in Hu (2024). The flux was calculated according to the WPL-corrected method (Webb et al., 1980) and subsequently narrowed down to a frequency of every half hour. Observations were made continuously from October 24 to December 21, 2021 and from August 15 to September 13, 2022. The flux observations obtained from eddy covariance observations are

specifically used to fit the coefficients of the mass conservation model at the date and time they are available, providing a ground-truth at the place and time the observations were made.

### 2.2 Mass conservation equation

    The method used in this study is based on the continuity equation for conservation of mass of chemical substances in the atmosphere.

$$E = \frac{d}{dt}XCH_4 + H + D + T \tag{1}$$

Where E is the emission flux, $\frac{d}{dt}XCH_4$ is the time derivative of XCH$_4$, H is the chemical gain or loss of CH$_4$, D is the deposition of CH$_4$, and T is the transport gain or loss of CH$_4$ on each given grid in space and time. Due to the slow nature of CH$_4$ loss induced by OH, its low solubility, and its slow removal due to other oxidants and bacteria as compared to the daily-scale

observations used herein, the terms for chemical gain H and deposition D in Equation (1) are approximately zero, allowing Equation (1) to be simplified as:

$$\frac{d}{dt}XCH_4 = E - T = E - \alpha * \left(\nabla \cdot (XCH_4 * U)\right) - \beta * \nabla \cdot (\nabla XCH_4) \tag{2}$$

The transport term T is approximated by the combination of advection and pressure induced transport $\nabla \cdot (XCH_4 * U)$ and diffusion $\nabla \cdot (\nabla XCH_4)$, where U is wind field. We specifically use the methane flux measured at Changzhi as E and calculate the time derivative, transport, and diffusion terms for the corresponding position and time over Changzhi. The coefficients α and β are then calculated using Multiple Linear Regression. The resulting emissions have been previously demonstrated to work well in Shanxi (Hu et al., 2024). In this work, the trained model is used to calculate emissions across all grids on all days where there is sufficient available TROPOMI and meteorological data to compute all of the terms. The uncertainty range of emissions is determined by the range from the 20th percentile to the 80th percentile of the emission values obtained through multiple sets of coefficients.

## 2.3 Observational Uncertainty Analysis

Our model and other more simplified approaches (Cohen and Prinn 2011; Liu, M et al., 2021; Yu et al., 2023) rely upon concentration gradients, such as the transport term and diffusion term in our equations and the divergence method used by others(Liu M et al., 2021; Veefkind et al., 2023) , when considering a certain volume of air, we can calculate the number of $CH_4$ molecules flowing into and out of that volume based on column density and representative wind speed and direction acting upon that atmospheric column. Under steady state conditions, a positive difference indicates the presence of a $CH_4$ emission source. However, the value of a gradient does not linearly vary with the uncertainty in the observations. For instance, if the XCH4 values at two adjacent grid points are 1800 and 1900 ppb, with a 10% uncertainty, they can range from 1620-1980 ppb and 1710-2190 ppb respectively, leading to a possibility that the gradients could range from negative, to zero, or even moderately positive. Sinks comprise chemical losses, such as reaction with the hydroxyl radical, OH, and atomic chlorine and biological losses, primarily uptake by methanotrophic microbes in soil. The lifetime with respect to reaction with OH in the troposphere is between 10 and 11 years and the average lifetime of methane in the atmosphere is 9 years, methanotrophic bacteria achieved methane oxidation rates of about 5.6 nmol/$cm^3$/day. Forest soils that cover about 30% of the Earth's land surface absorb about 26 to 34 Tg $CH_4$ per year. However, we calculated TROPOMI XCH4 data day by day, grid by grid, and the two cases have little effect on methane concentration. Hence, there is no physical means by which a negative lifetime can occur in our framework. Therefore, only white noise generated by the nonlinearity of the concentration gradient is possible to achieve negative emissions.

However, merely removing negative values is not mathematically consistent with white noise, due to the randomness of the uncertainties applied to the observed data being equally likely to be positive or negative. Therefore, any values which fall in the probability distribution which is best fit by a normal distribution with a center at zero, are associated with this noise. They are computed values, but are just due to the noise associated with the observational uncertainty. It is essential to compute

how these uncertainties impact the real conditions when considering the spatial change in the observed fields. This includes not only the observed values themselves, but also whether or not the underlying assumptions of the model used are still relevant, or whether the simple idea of a plume model may still be possible. For example, if the gradient switches direction when uncertainty is applied, the chances of the plume being real are significantly reduced, with the original detected plume more likely being just noise, rather than a real signal.

To analyze this effect, we use the mass conservation equation to quantify methane emissions in the area around the Waliguan (WLG) long-term base station, because there are no large known sources of methane emissions in this area, the effect of noise due to the non-linearity of the gradient term can be better demonstrated, with most recent studies computing emissions using background subtraction (Schneising et al., 2020; Liu, M et al., 2021; Veefkind et al., 2023) which instead of computing actual emissions rather computes a signal based on observational noise in both the background value and the enhancement value in addition to the effects of any actual emissions. Instead, this work subsequently uses the entire column loading XCH$_4$ and applies two different magnitudes of random perturbations to the observations: 10% and 20%. In each case, a respective value ranging from -10% to +10% (or 20% respectively) is made independently to each TROPOMI retrieved XCH$_4$ data point, day-by-day, and grid-by-grid to simulate realistic and unbiased TROPOMI retrieval errors. 1000 unique sets of perturbations have been performed. Each of these 1000 data sets have been used to retrain the emissions via equation (2), with the resulting average value grid-by-grid and day-by-day used.

**2.4 Threshold retrieval**

To identify potential emission sources in the study area and separate these from white noise, we take a two-step approach.

Spatial filtering: we filter all grids in terms of their mean value, with any grids having temporal average emissions values smaller than the 65$^{th}$ percentile of emissions (in this case 26.5 μg/m²/s) removed, to account for the fact that these grids likely are contributed to exclusively by white noise. Sensitivity tests are performed using different cutoff values for the first cutoff, specifically the 50$^{th}$ percentile of emissions (12.4 μg/m²/s), and re-computing the resulting emissions each time using all of the TROPOMI uncertainty levels as used elsewhere.

Temporal filtering: the idea is extended to the temporal domain, considering the seasonal variations in emissions (Varon et al., 2025), we conduct monthly filtration, specifically assuming that any negative emissions values must be due to observational uncertainty, and therefore any positive value of the same magnitude or smaller on that same grid is also due to uncertainty. Specifically, within each remaining grid point, the most negative emission value computed is identified -θ μg/m²/s, and all values in the range from [-θ: θ] μg/m²/s are then excluded. This filter is applied grid-by-grid basis, since the uncertainty is assumed to be an intrinsic property of each grid point, consistent with how the retrievals are computed.

To further validate our method, we applied it to the Permian Basin, comparing our results with strong CH$_4$ point sources mapped and quantified by Cusworth et al. (2021) using airborne imaging spectrometer technology

# 3 Results and discussion

## 3.1 Observation error

Due to the following factors, retrievals of $CH_4$ from TROPOMI may have relatively large uncertainties in some regions
(noting that their influence varies as a function of location and time). First, when there is overlap between $CH_4$ and other species which are not resolved (i.e., waveband resolution of the sensor). Second, when atmospheric conditions reduce incoming solar radiation in the wavebands relevant to $CH_4$ retrieval (i.e., cloud optical depth and AOD). Third, when atmospheric conditions alter the band-by-band properties of incoming and scattered solar radiation in the wavebands relevant to $CH_4$ (i.e., BC and CO). Fourth, when changes in the land-surface properties have occurred which are not properly modeled or included,
such as due to development, greening, industrial growth, agriculture changes, etc. (i.e., changes in surface reflectivity).

The uncertainty of the aerosol products of TROPOMI is 3% to 5% (Torres et al., 2020; Tiwari et al., 2023). As a first order problem, no gas retrievals can be performed until the surface albedo, clouds, and aerosols are first considered (Balasus et al., 2023), or the combination is considered in tandem (Chen et al., 2022). In addition to this point, it has been demonstrated that TROPOMI L2 $XCH_4$ retrievals are routinely underestimated over regions with high aerosol loadings (K. Li et al., 2024).
Therefore, since the region considered has a variable AOD, and one which is quite different from the traditional datasets employed to initiate the inversion algorithm (Liu, Z et al., 2024), therefore this factor must also be considered in the area being analyzed. For this reason alone, it is not possible for the $CH_4$ retrieval to have an uncertainty lower than the aerosol retrieval uncertainty. TROPOMI $NO_2$ has an error range from 10% to 40% or more (Boersma et al., 2018; Pollard et al., 2022). It is also known that TROPOMI CO has an error range of at least 10% to 20% (Sha et al., 2021). The uncertainties in the traditional
datasets employed with respect to CO over China are significant (Li et al., 2025; Wang et al., 2025). This is important since the CO retrieval is made at the same waveband as the $CH_4$ retrieval, and therefore any uncertainty in one product will yield an uncertainty in the other product as previously identified (Gaubert et al., 2016, 2017).

The region around Waliguan was specifically selected due to its long-term daily surface observations of $CH_4$ from 1992 through 2024 (Zhou et al., 2004; Liu, S et al., 2021). Waliguan is the world's highest atmospheric baseline monitoring station
(at 3810 meters) and the only one in the interior of the Northern Hemisphere, making it the best possible representation of a clean, continental, high atmospheric station in mid-latitudes, and best represents the world's middle atmospheric $CH_4$ levels. This point is critical, since an outsized number of coal mines with high emissions from China are similarly uniquely located in Shanxi, Shaanxi, Inner Mongolia, Xinjiang, and other places located at high elevations and far away from the sea (Yang et al., 2023; Zhang et al., 2020), and is more representative of the conditions under which $CH_4$ is emitted in China, as well as
some other lesser studied regions around the world (Sadavarte et al., 2021; Hancock et al., 2024). A direct comparison between the surface observations and the TROPOMI $XCH_4$ column concentration, as clearly observed in Figure 1b, demonstrates a difference up to 176ppb, or a 9.4% error. Observed vertical $CH_4$ profiles were made by Tao et al. (2024) Figure 2 using AirCore at the same time and very close to where this work's satellite observations were analyzed. The results compare closely with the surface observations made at the WMO station in Waliguan, validating that the surface observations in this region are a

reasonable representation of the column average values, as shown in Figure 1. Based on this, we add a random perturbation to each retrieved value of TROPOMI XCH4 in the range from -10% to 10% to simulate the uncertainty in the retrieval data itself. This is also consistent with the observational study of Frankenberg et al. (2016), indicating that the observed error can range over 20%.

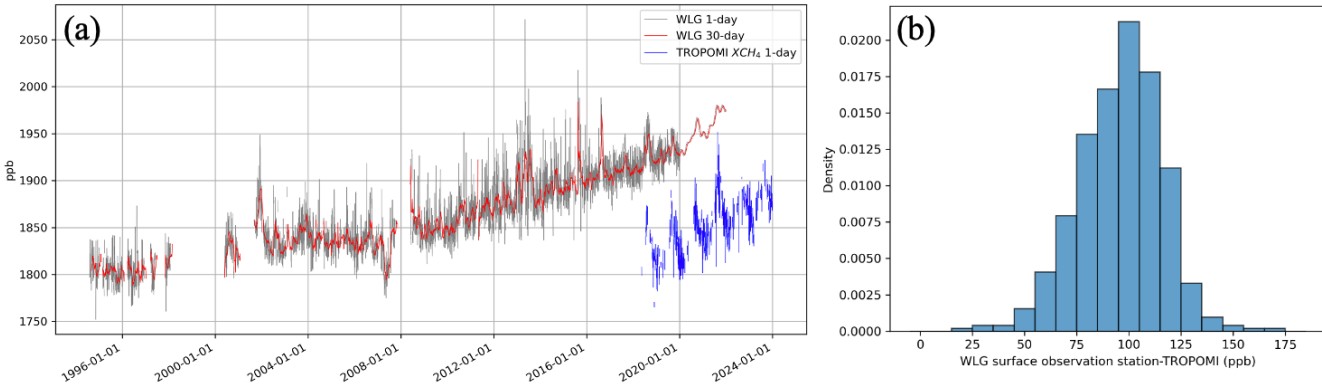

**Figure 1: (a) Daily (grey) and 30-day mean (red) methane concentration data from Waliguan from August 1994 to December 2021 and daily series of TROPOMI XCH4 (blue) from April 2018 to December 2023; (b) The difference between Waliguan daily methane concentration and TROPOMI XCH4.**

### 3.2 Comparison of emissions before and after application of filtration in Waliguan area

As illustrated in Figure 2, the distribution of computed emission over all pixels on all days exhibits a predominantly
normal distribution pattern, and therefore closely resembling the distribution of white noise. Upon closer inspection, the distribution reveals obvious positive bias, signifying the presence of mixture of genuine subset of emissions values which are computed not due to the observational uncertainty contained within the overall set of computed emissions data. The distribution derived without employing the two-step filtering process, but incorporating a 10% TROPOMI uncertainty, shows only marginal deviations from the original emissions.

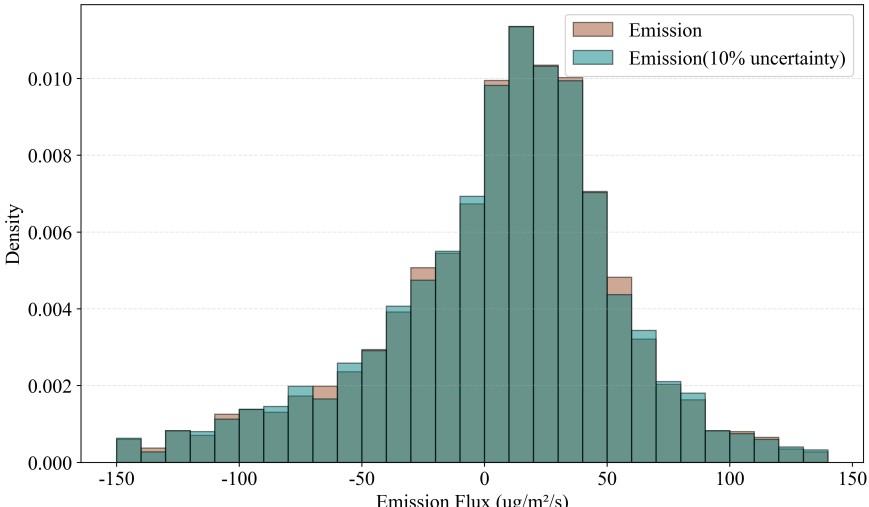


**Figure 2: PDF of methane emissions from the Waliguan region calculated using TROPOMI XCH$_4$ and XCH$_4$±10% based on the mass conservation model**

Figure 3a highlights a significant occurrence of near-zero or negative emissions in spatial regions where no emission sources are expected, suggesting the presence of numerous false positives. To address this, a probability density function (PDF)

threshold is established based on the point where the distribution deviates from normality on the positive side. Grids with mean emissions exceeding this threshold are retained to distinguish potential real emissions from noise-dominated grids. An example of such a grid, marked by a black box in Figure 3a, demonstrates a mean emission value surpassing the threshold.

Following this initial spatial filtration, a secondary temporal filtering step is applied to the remaining pixels, Figure 3b and c show the time series before and after filtering by the red grid in Figure 3a. This step effectively eliminates unphysical

values and those arising solely from observational noise, ensuring that only robust emission estimates are considered for further analysis.

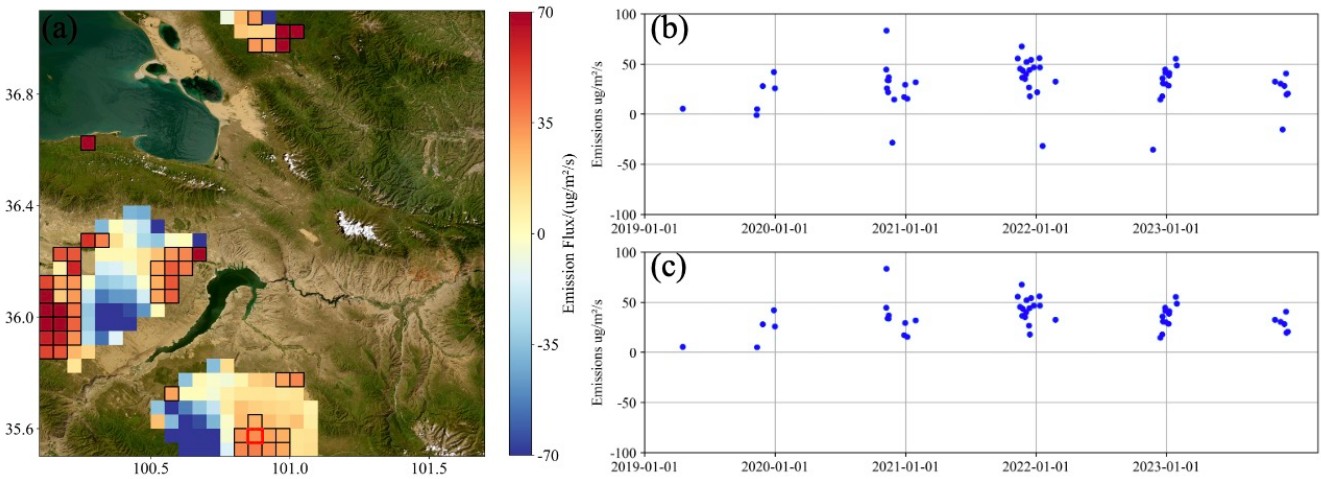

**Figure 3: (a) Five-year emission mean calculated by TROPOMI XCH$_4$, the mean of the grids in the black boxes are greater than the set threshold (12.4 μg/m$^2$/s) (background portion of image is from Esri World Imagery); (b) and (c) are the unfiltered and filtered emission time series of the red grid in (a).**

The spatial distribution of day-to-day and grid-to-grid emissions over five years is presented in Figure 4a-c (pre-filtering) and Figure 4d-f (post-filtering). Negative emissions are mainly concentrated in areas A and C of Figure 4a. The point of the filtering is to determine whether or not observational uncertainty or noise was the source of the retrieved emissions, or if the retrieved emissions were due to a physical signal. Some physical factors which contribute to signal observational uncertainty or noise which still exist in the QA>0.5 data include but do not fully filter for thin clouds and aerosol layers as well as moderate variations in water vapor, or medium-low albedo (Hu et al., 2016). This QA level also does not consider aerosol absorption or apply any constraints on co-absorbing gasses including but not limited to carbon monoxide. However, this work specifically analyzed a some of these drivers, and clearly determined that some of these drivers in fact contributed to those grids which were filtered (i.e., areas in which the emission derived from the signal were negative or sufficiently small or large and positive to be considered noise).

Aerosols impact XCH$_4$ in two different ways, through scattering and through absorption. First, aerosols increase radiative scattering, changing the entirety of the stream of energy (Kahn et al., 2023; Liu et al., 2024; Tiwari et al., 2023, 2025), while also absorbing radiation, affecting line-by-line radiance absorption used to invert XCH$_4$ based on Beer's Law, DOAS, or similar physics-based techniques (Kuhlmann et al., 2025; Tian et al., 2021; Guan et al., 2025). We have included observations of AOD at a band as close to that retrieved from TROPOMI as possible (specifically observed by MISR at 865nm). As shown in Figure S1h, almost all of our valid emissions occur at locations where the 2019-2021 average AOD is less than 0.3. Figures 5b and e show the PDFs of AOD corresponding to the points in space and time of the invalid and valid emissions, respectively, while spatial plots (see Figure S2b and e) detail that while all grids are low, that those grids with valid emissions have lower AOD than grids with invalid emissions. Furthermore, we have analyzed the TROPOMI AOD_SWIR product (Figure S2), and found similarly that where our emissions are valid, the AOD remains both very low, as well as being smaller than the AOD on the invalid emissions locations, as show in Figures S2a and b.

Next, we examine the impact of aerosol absorption based on MISR AAOD at 443nm, 865nm, and the ratio of the two AAODs as given in (Figures S1e, f, g) following Liu, Z et al. (2025). Areas with a large AAOD ratio indicate the presence of larger-sized absorbing aerosol, which is more likely to impact the radiation wavebands used in the inversion of XCH$_4$. We observe that the ratio is somewhat larger over areas with emissions we have filtered (such as area A in Figure 4a). Furthermore, we observe that most of the grids with valid emissions are located are in areas with very low AAOD ratios. The PDFs of the AAOD ratio for the invalid and valid emission points, shown in Figure 5a and d respectively. Therefore, our approach is successful in determining that the pixels more impacted by aerosols are in fact filtered.

The surface albedo in this region as shown in Figure S1d. Our retained emissions do not occur at locations with either very low or very high surface albedo. Moreover, compared with the retained emissions (Figures 5c and f), the surface albedo corresponding to the filtered emissions are typically found closer to the extreme ends of the albedo range.

These factors alone cannot explain the total uncertainty, which may be influenced by other parameters, such as cloud, sensor and waveband resolution issues, etc. This result offers a physical explanation of why there is a substantial uncertainty in the XCH4 retrieved values. What is also important is that our approach is capable of detecting such a non-linear uncertainty propagation, while standard emissions estimation approaches (Schneising et al., 2020; Veefkind et al., 2023; Hancock et al., 2025) in fact compute negative emissions, without realizing that the emissions are merely due to observational uncertainty. To be clear, this approach herein is further validated, since applying the uncertainty in general does actually capture a subset of physical driving factors which are expected to lead to greater retrieval noise.

A comparison of the filtered emissions in Figure 4d with the actual geographical map reveals that the filtered results accurately capture known emission sources: towns and villages. The emission time series (Figure S3) in regions A, B and C show that emissions are mainly concentrated around the Spring Festival, which is consistent with the increase in methane emissions caused by the short-term influx of people and increase in human activities during this period. However, these small-scale anthropogenic sources are omitted in existing emission inventories over this region (Crippa et al., 2024). Our filtering method effectively removes noise, retaining only grid points corresponding to plausible emission sources. All grid points identified as emissions in this study are scientifically justifiable, whereas many grid points initially classified as white noise (Figure 4a) lack any known anthropogenic or natural emission sources substantial enough to have an emission corresponding to the lowest calculated value herein.

The distributions in Figure 4b, e and Figure 4c, f correspond to the 10% and 20% TROPOMI uncertainty scenarios, respectively. These distributions largely overlap with the results shown in Figures 4a and d. Specifically, when the TROPOMI XCH4 data incorporates a 10% error, the filtered grid points align closely with the original emissions, with only a single grid discrepancy. This demonstrates the efficacy of our method in filtering out noise induced by the nonlinearity of the gradient term. However, when the TROPOMI error is increased to 20%, the robustness of the filtering method slightly diminishes. When the 50$^{th}$ percentile emission value (12.4 μg/m$^2$/s) of the emission result was used as the threshold for mean filtering in the first step of filtration (Figure S4), the grid discrepancy in regions A and B was not obvious, but emissions occurred in the desert area of region C, this indicates the necessity of the first step of spatial filtering. Overall, Equation (2) successfully conserves mass by balancing the nonlinearity in the gradient terms through the contributions of other terms, ensuring reliable and consistent results.

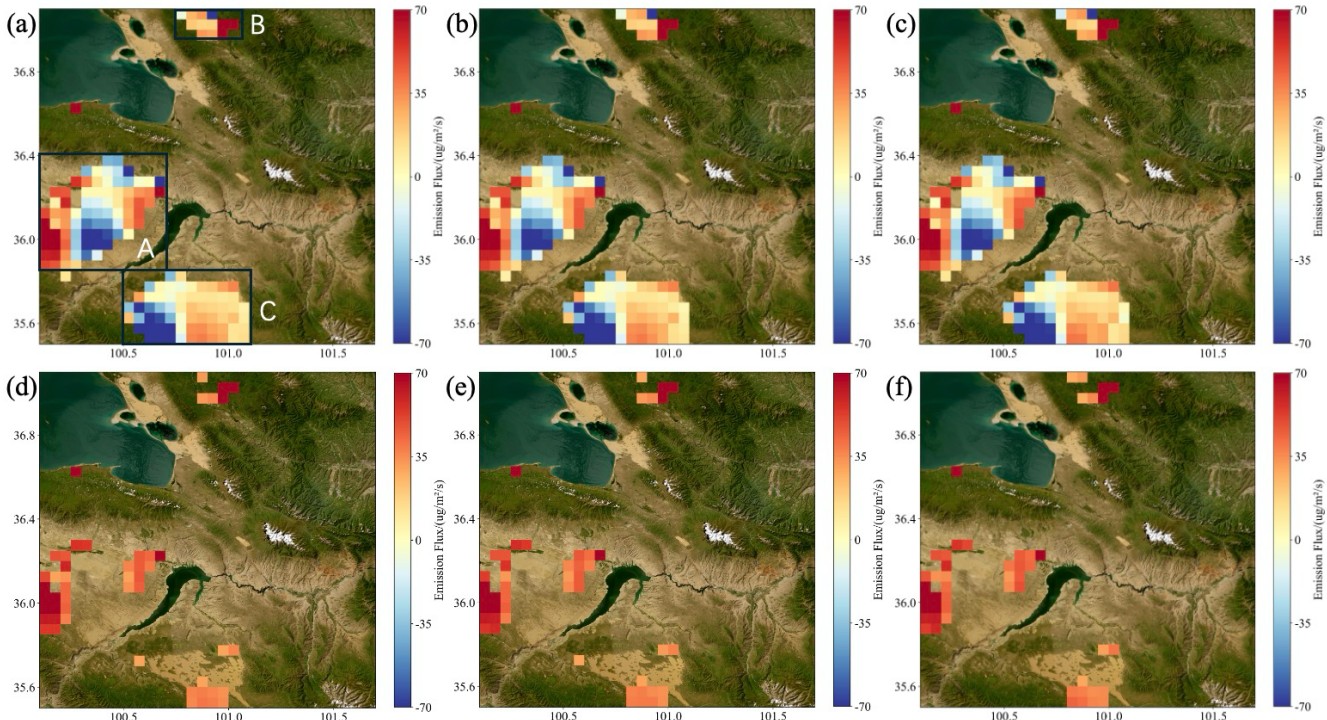

**Figure 4: (a), (b), and (c) are maps of five-year emission averages calculated using TROPOMI XCH₄, XCH₄±10%, and XCH₄±20% respectively; (d), (e) and (f) are maps of the five-year emission average obtained after the two-step filtering is applied corresponding to (a), (b), (c). All backgrounds are from Esri World Imagery.**

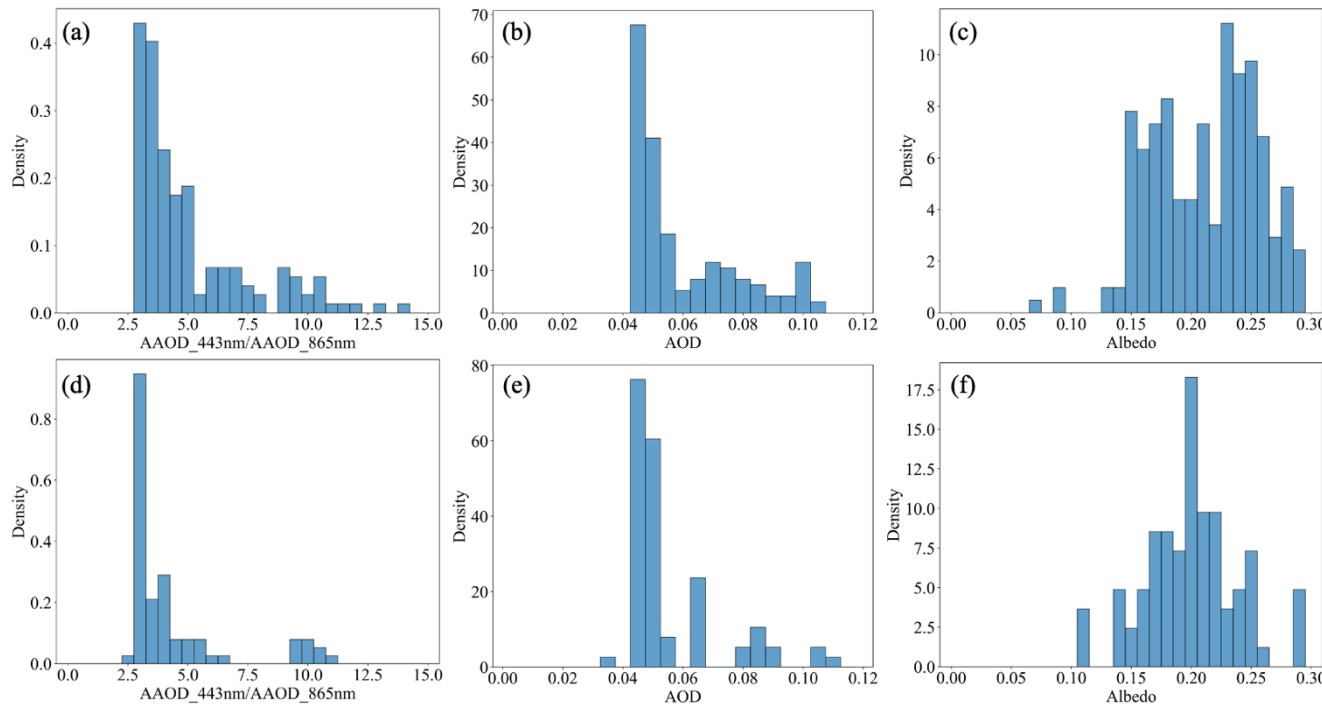

**Figure 5: (a), (b), and (c) are the PDFs of the ratio of MISR AAOD observed at 443nm to AAOD observed at 865nm, MISR AOD observed at 865nm, and albedo corresponding to the location of the filtered emission (invalid emissions), respectively; while (d), (e), and (f) are the same respective values, but corresponding to the location of the retained emissions (valid emissions), respectively.**

Scatter plots in Figures 6a and b compare the emissions of 0% TROPOMI uncertainty with those of 10% TROPOMI uncertainty and 20% TROPOMI uncertainty respectively, with the respective $R^2$ and RMSE being 0.99, 3.41 and 0.98, 6.81. The unfiltered emissions contain points which may be mathematically correct, but physically unreasonable including points with a roughly zero mean and substantial noise, or random extreme values. The results after two-step filtering are given respectively in Figures 6c and d, and have respective $R^2$, RMSE of 0.99, 3.7, and 0.95, 7.37. Although the number of emissions after filtering in Figures 6c and d decreased by 70% compared to Figures 6a and b, $R^2$ was similar at 10% TROPOMI uncertainty case, with RMSE increasing by 0.29, and a slight reduction in the $R^2$ in the 20% TROPOMI uncertainty case, with RMSE increasing by 0.56, which actually indicate that overfitting is less of an issue, consistent with the fact that the observations contain actual uncertainty.

The range of filtered emissions assuming 0%, 10%, and 20% TROPOMI uncertainty bounds respectively are 0.3-187 μg/m²/s, 0.2-187 μg/m²/s, and 0.5-195 μg/m²/s. In all cases applying the methods herein led to all negative values (unphysical) being filtered, as well as the largest positive values also being filtered. This is consistent with the fact that some very high observed values are also noise, and that the process herein in fact applies an unbiased and reasonable result. These discrepancies in the mean values arise because the first and second order gradient terms both behave non-linearly when observational

330   uncertainties are included. The fact that the linear terms and the temporal derivative are able to buffer the non-linearity allows the resulting emissions to be robust, and requires that emissions calculations must include these terms to be stable.

Many studies after calculating emission either use an absolute value of the gradient or simply remove negative resulting values, either way retaining only positive results (Veefkind et al., 2023; Liu, M et al., 2021). We have performed the same analysis approach and show the results in Figures 6e and f to compare the base emissions results against the 10% and 20%

335   TROPOMI uncertainty cases respectively. This method results in emissions that include a large amount of noise both near zero as well as some extremely high values at the top. Furthermore, it is found that these extremely high and nearly zero values occur in locations without any known emissions sources. This increase in near-zero noise leads to genuine small emissions sources being indistinguishable from noise, resulting in a lower average emissions per grid, and a far larger number of grids than is realistic.

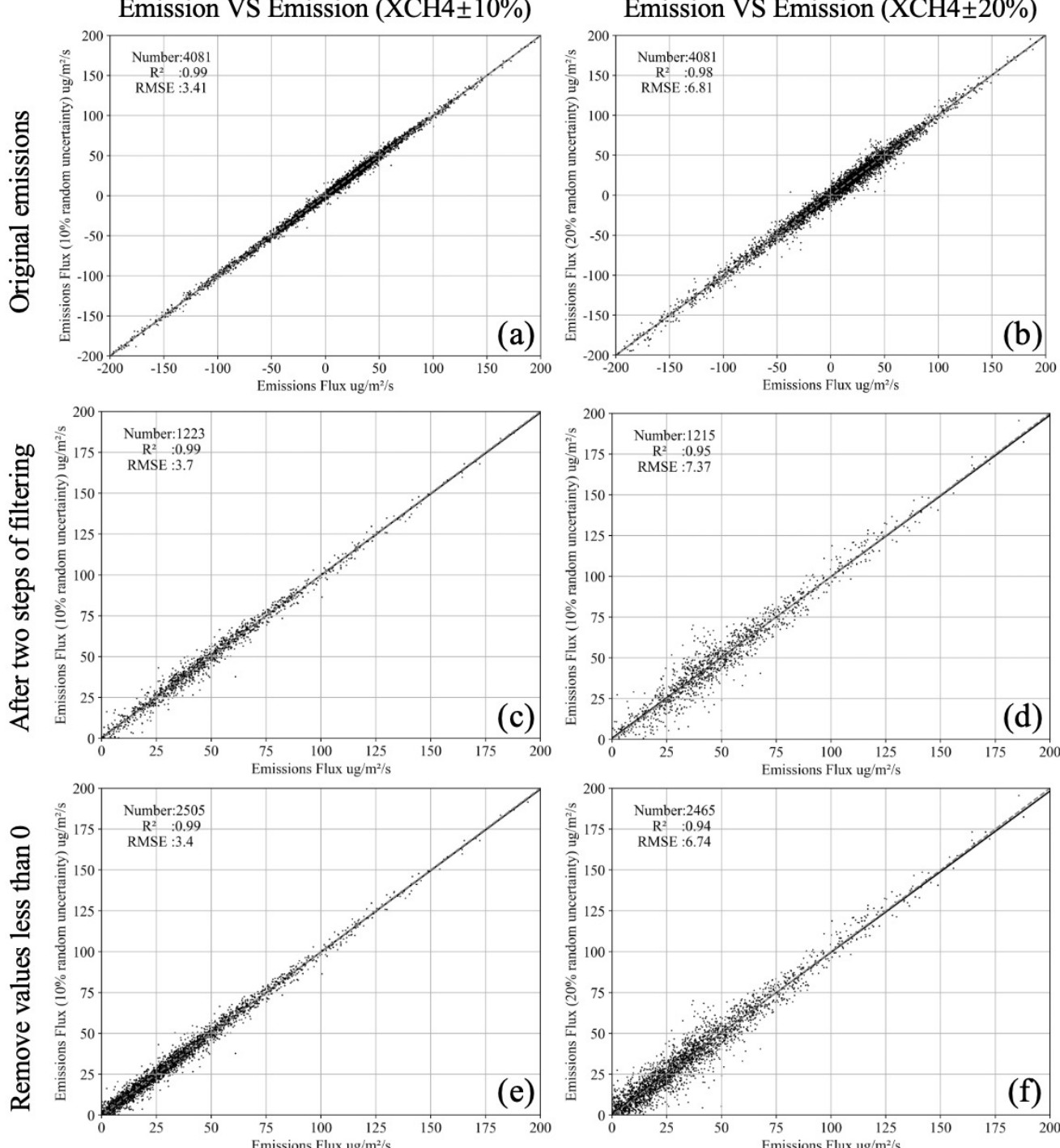

**Figure 6: (a), (c), (e) Scatter plots representing each computed emissions value in terms of grid-by-grid and day-by-day from TROPOMI XCH$_4$ (x-axis) and TROPOMI XCH$_4$±10% (y-axis); (b), (d), (f) are as (a), (c), (e) respectively, but where the y-axis is TROPOMI XCH$_4$±20%. Plots (a) and (b) contain all computed emissions before any filtration. Plots (c) and (d) show the results after both thresholds are applied. Plots (e) and (f) apply the traditional method of removing all computed emissions lower than 0 μg/m²/s.**

### 3.3 Methane emissions of the Permian Basin

The Permian Basin, situated in western Texas and southeastern New Mexico, is the largest and fastest-growing oil and gas-producing basin in the United States. According to the U.S. Energy Information Administration (EIA), oil and natural gas production in the Permian Basin surged by 390 percent and 350 percent respectively, from 2014 to 2019. By the end of 2020, the basin accounted for approximately 38 percent of total U.S. oil production and 17 percent of total U.S. natural gas production. To evaluate the applicability and robustness of our method, we applied the mass conservation equation to quantify methane emissions in this region from 2019 to 2020.

As depicted in Figure 7a, negative and near-zero values appear in certain areas, with similar results observed when applying divergence-based methods to quantify emissions in this region using TROPOMI with both plume-based methods (Veefkind et al., 2023) and Gaussian integral method (Schneising et al., 2020). The unfiltered computation of grid-by-grid and day-by-day emissions exhibits a significant positive shift compared to Waliguan, a reasonable outcome given the presence of genuine and substantial emission signals, Interestingly, the overall result still contains a large amount of data centered around 0, as evident in Figure 7b. As previously demonstrated, traditional methods that simply remove negative values or employ background subtraction retain substantial noise near zero as well as some maxima, leading to artificially low emission estimates.

The results after applying our two-step filtration method are presented in Figure 7c. Post filtration, the mean emission values have increased significantly compared to the pre-filtration values, with all negative values and noise near zero effectively removed. The quantified true emissions range from 0.5 to 210 μg/m²/s, with the range of values slightly lower than but overlapping with results from other studies such as from high gas coal mining areas in Shanxi China (Hu et al., 2024; Qin et al., 2023). Notably, unlike background subtraction, our filtering method does not indiscriminately remove all values below a predetermined threshold. As illustrated in Figure 7d, the minimum emission value detected after filtering is 0.5 μg/m²/s, consistent with the smallest detectable signal of 0.2–0.3 μg/m²/s observed in the Waliguan region, demonstrating the capability of our method to identify small emission signals.

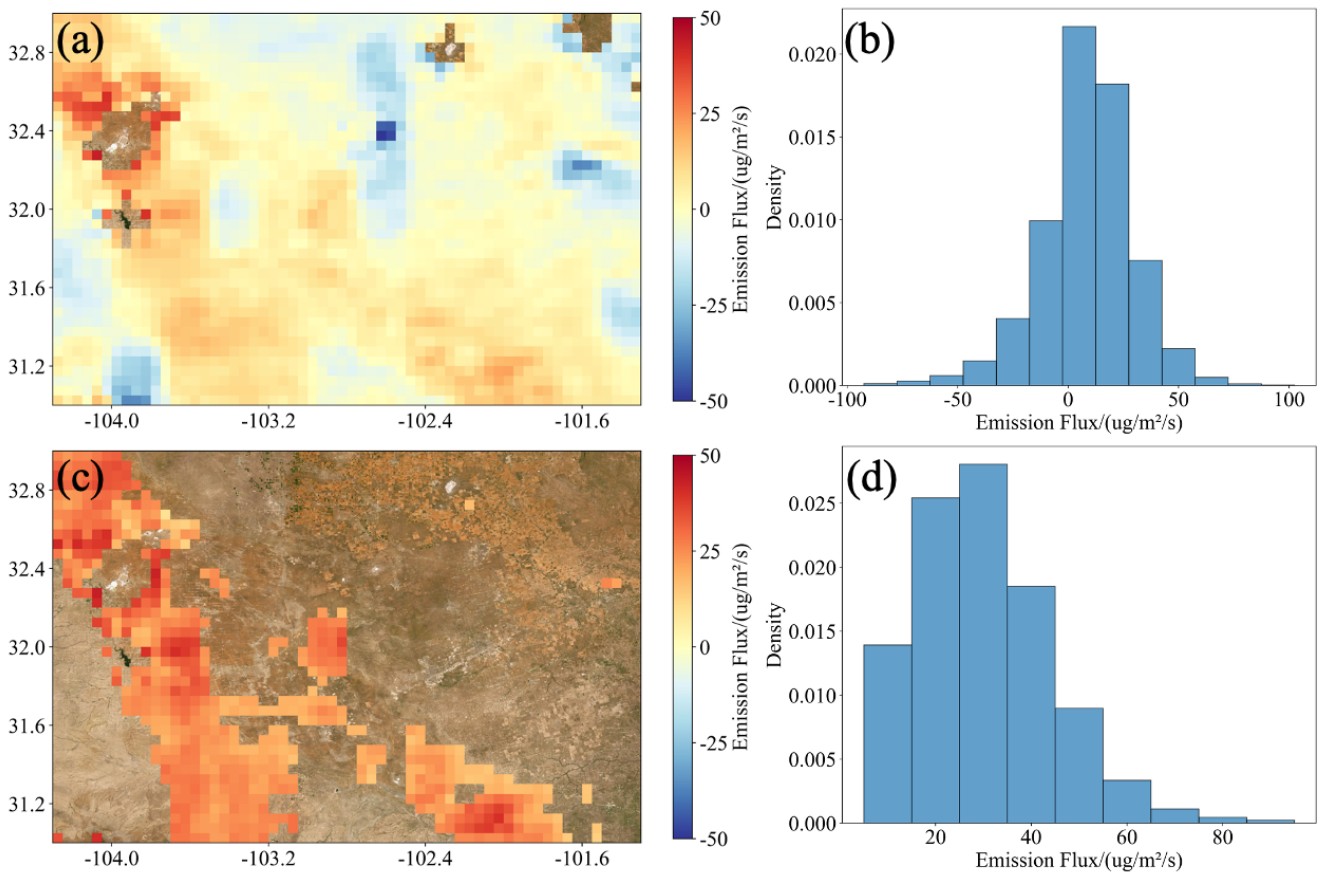


**Figure 7: (a) Annual mean methane emission fluxes from the Permian Basin for 2019-2020, (b) PDF of the Permian Basin emission fluxes for 2019-2020, (c) Annual mean methane emission fluxes from the Permian Basin after filtration in 2019-2020, (d) PDF of the 2019-2020 methane emission fluxes from the Permian Basin after filtration. Backgrounds of (a) and (c) are from Esri World Imagery.**

To better compare with other inventories many of which use emissions in terms of mass per time per grid, we first convert

emissions fluxes (μg/m²/s) into emission rates over each entire TROPOMI grid (kg/h/grid). The EDGAR emission inventory over the Permian Basin is shown in Figure S6a, and Figure S6b shows the difference between our emission results and EDGAR, with the grid-by-grid ranging from -285 to 3830 kg/h/grid, and the 95th percentiles of the difference is 2920 kg/h/grid. The EDGAR emission inventory is significantly lower than our results, in part due to many emissions grids missing from their dataset, although their grid with the highest emission is still lower than our result, indicating that our approach does not have

a high bias. We also compared the differences between the inventory of Cusworth et al. (2021) and EDGAR, as shown in Figure S6c. The difference range is from -3240 to 8440 kg/h/grid, and the 95th percentiles of the difference is 2450 kg/h/grid, which is close to the difference range between our results and EDGAR.

Cusworth et al. (2021) measured the emission rates of individual sources, and due to the intermittent nature of the aircraft observations, the number of days with detectable emissions varied from 0 to 12 days for each source, with the vast majority

having only 1 or 2 days of data. When multiple emission sources were located within a single grid, their emission rates were summed to represent the total emission rate of that grid (Figure S7). Figure 8a shows our filtered emission mean, overlapping with the grid of ground emission sources. It's clear that our results are larger, which is consistent with the observations made by aircraft having scan widths of 3 and 4.5km, which is always smaller than our grid resolution of 0.05 by 0.05, which is about 5km. This means that even if their scan crossed the center of our grid, our grid would still contain information outside of their

scan width, and if the scan only crossed a small amount of our grid, then the grid would contain far more information.

The red dots in the scatter plot of Figure 8b represent the points where our emission results overlap with Cusworth et al. (2021) in both time and space, with R=0.8 and MAE=660 kg/h/grid, both indicating a reasonable agreement. The black dots and blue dots respectively represent our effective emissions and the emission estimates of Cusworth et al. (2021) on the same grid, but during other days when Cusworth et al. (2021) does not have observations but TROPOMI does. We found that the methane

emissions from these emission facilities varied significantly over time, indicating that TROPOMI allows for a better understanding of how emissions dynamics may change over time.

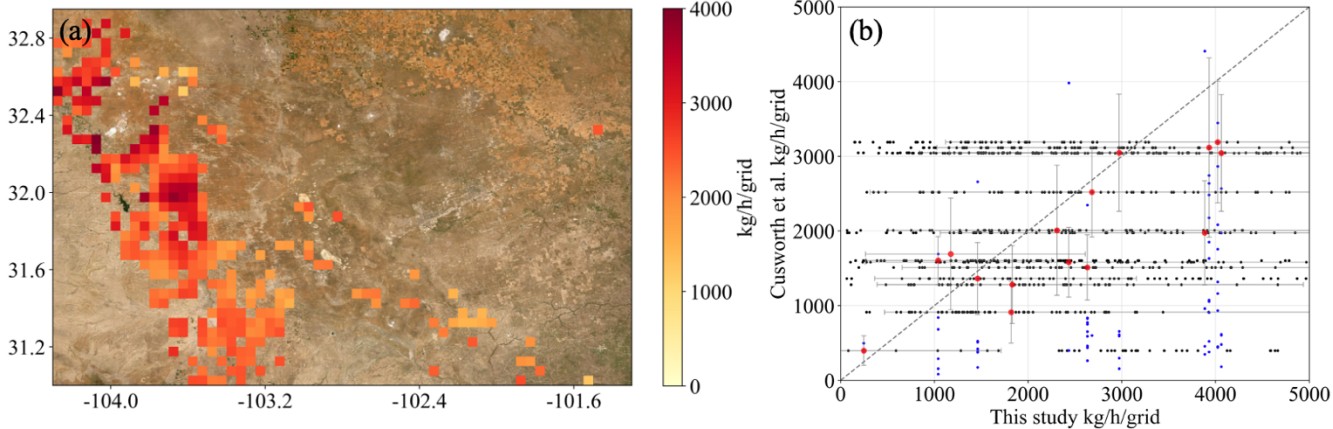

**Figure 8: (a) Annual mean CH₄ emissions (kg/h/grid) on grids which overlap spatially with an emission observed by observed by Cusworth et al. (2021); (b) The red dots in the scatter plot represent the points where our valid emissions**

**grids overlap in space and time with those of Cusworth et al. (2021). The black and blue dots denote emission estimates from our study on the same grids as observed by Cusworth et al. (2021), but observed by TROPOMI on different days, representing temporal variation. The gray error bars represent the associated uncertainties. The background of (a) is from Esri World Imagery.**

**4.Conclusion**

This work used a mass conserving partial differential equation, with terms trained based on observations of CH₄ emissions from Changzhi, Shanxi, to compute emissions of CH₄ using daily and grid-by-grid TROPOMI XCH₄, explicit uncertainty of retrieved XCH₄, and reanalysis meteorological data, over and around the relatively clean area around the long-term WMO CH₄ background observation station at Waliguan and the heavily emitting CH₄ region of the Permian Basin. Inclusion of uncertainty

in retrieved TROPOMI XCH$_4$ values are used to recompute emissions in a manner that considers the physically relevant non-linear effects of gradient calculation. The work further introduced an unbiased two-step filtering method, which effectively removes all negative values as well as both small and larger uncertain values in an unbiased manner, yielding results which are realistic and match well with underlying driving factors.

In Waliguan region we have identified CH$_4$ emissions, with a range from 0.3 to 187 μg/m$^2$/s, and validated that all resulting points are all locations containing known anthropogenic CH$_4$ source activities. The minimum inverted emission value ranges from 0.4 to 0.5 μg/m$^2$/s, depending on the amount of TROPOMI uncertainty applied. Due to our model's consideration of both the wind times concentration gradient and second order concentration gradient terms and actively propagates the TROPOMI data uncertainty, requiring a buffering from the linear and temporal derivative terms, a factor overlooked either in the name of model simplification or over-stiffness. When following the typical approach of removing negative emissions or using absolute values, excessive noise near 0 μg/m$^2$/s obscures the true signal, while noisy but large values allow points to enter into the solution space which are also too large to be realistic.

To further validate our approach and results, we compared our emission rates computed over the Permian Basis with aircraft imaging spectrometers estimated by Cusworth et al. (2021). Our observations span a longer time series than Cusworth et al. (2021) (more temporally representative) and that the per-grid area is also larger than Cusworth et al. (2021) (per grid possibility of missed sources. This highlights the robustness of our method in capturing both smaller than currently considered possible emissions signals using TROPOMI, as well as reducing some amount of very large emission signals, which likely are a result of TROPOMI inversion noise on the positive side.

This work identifies three weaknesses to be addressed by future work. First, there is need to more carefully consider the inversions of multiple retrieved species in tandem, not just a single species, since other species may impact the inversion of CH$_4$. Second, future satellite missions which are capable of retrieving XCH$_4$ at a slightly higher spatial resolution than TROPOMI may yield additional gains. Third, the use of additional platforms that can yield more precise XCH$_4$ calculations will improve the ability to detect not only more signals, but to widen the possible detection range. Furthermore, there is the issue that the work herein is not reproducing a large number of very small emissions values, as claimed by some works (Chen et al., 2025), although the results seem to have a lower detection threshold than other TROPOMI based works (Cusworth et al., 2022).

In summary, our method provides a reliable framework for quantifying CH$_4$ emissions by effectively distinguishing genuine emission signals from noise, rigorously accounting for observational uncertainties, and validating results against independent datasets. This represents a significant advancement over traditional approaches, enabling more precise identification and quantification of CH$_4$ sources. Importantly, studies relying on TROPOMI or other satellite data that fail to actively incorporate observational uncertainties often underestimate true emissions on a grid-by-grid basis while simultaneously overestimating the number of emitting grids. This underscores the critical need to integrate uncertainty propagation into emission quantification methodologies to achieve robust and reliable results.

**Data Availability**

TROPOMI $CH_4$ product (v2.4) can be found here: https://s5phub.copernicus.eu/dhus/#/home, last access May 2021(Landgraf
et al., 2024). Daily ground station $CH_4$ data for Waliguan are obtained World Data Centre for Greenhouse: https://gaw.kishou.go.jp/search/file/0013-2015-1002-01-01-9999 (last access: 10 October 2025). The wind Gases data were obtained from the European Centre for Medium-Range Weather Forecasts ERA-5 reanalysis product (Hersbach et al.,2023). The ground measurement data and coefficients of Changzhi coal mine were obtained from Hu (2024). The point coordinates and emission rates of surface emission sources in the Permian Basin were obtained from Cusworth et al. (2021). The AAOD
product used in this study is from (Multi-angle Imaging SpectroRadiometer) MISR satellite observations and are obtained from the NASA Langley Atmospheric Science Data Center (https://doi.org/10.5067/Terra/MISR/MIL3MAEN_L3.004, NASA/LARC/SD/ASDC, 2008). The satellite base maps used in this work are all from Esri World Imagery. All emissions computed in this work are available for download at https://figshare.com/s/058f7f73953264e0d439 (Zheng et al., 2025).

**Authors contributions**

This work was conceptualized by JBC and BZ. The methods were developed by JBC and KQ. LL, WH, PT, SL, AG and HS provided insights on methodology. Investigation was done by BZ, JBC and KQ. Visualizations were made by BZ and JBC. Writing of the original draft was done by BZ and JBC. Writing at the review and editing stages were done by BZ and JBC.

**Competing interests**

The authors declare that they have no conflict of interest.

**Acknowledgments**

This study was funded by the International Science and Technology Cooperation Program of Jiangsu Province (BZ2024060). The satellite base map used in this study is from Esri World Imagery. These images are used strictly for academic, non-commercial research, in accordance with the fair use policy and in accordance with Eris' Terms of Service.

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
