# Peer review of "How can we trust TROPOMI based Methane Emissions Estimation: Calculating Emissions over Unidentified Source Regions"

_EGUsphere, 2025_

## Referee Comment (RC3)

How can we trust TROPOMI based Methane Emissions Estimation: Calculating Emissions over Unidentified Source Regions

The author presents a methodology for estimating methane emissions using TROPOMI data while accounting for observational uncertainty. These uncertainties are crucial, as they can significantly influence emission estimates. However, there are several critical aspects of the methodology that need to be carefully reviewed.

1. Selecting pixels with QA > 0.5 may introduce artefacts, potentially increasing uncertainty in the emission estimates. Rather than relying solely on this threshold, did you explore the use of AOD and albedo filters as recommended by Schuit et al. (2023) and Nesser et al. (2024)? If so, how did these alternative filtering approaches impact your results? I strongly recommend using only pixels with QA = 1.0, as lower QA values may include retrieval artefacts. Additionally, did you consider using the blended TROPOMI+GOSAT product, which reduces biases through the integration of GOSAT data? This could further enhance the robustness of your emission estimates.

2. Figure 2: Typically, pixels classified as water surfaces—especially those near coastal areas—are excluded due to their high uncertainty and potential for retrieval errors. Including TROPOMI pixels over water bodies can lead to unreliable or biased emission estimates, as these pixels often suffer from issues related to surface reflectivity and retrieval sensitivity.Additionally, the manuscript does not show a 2D map of TROPOMI $XCH_4$ in the Waliguan region and Permian Basin, which would help in evaluating the spatial context and data quality. It would also be beneficial to include maps of supporting variables such as albedo, AOD and surface pressure in this region to assess the robustness of the retrieval and filtering process. I recommend adding these visualizations and clarifying how water-body pixels were handled in your analysis.Canyou also add the TROPOMI observational density maps?

3. Lines 203–205: TROPOMI measures the total column-averaged dry-air mole fraction of methane ($XCH_4$), whereas the Waliguan station provides surface-level $CH_4$ concentrations. These are fundamentally different metrics, so differences between them should not be interpreted directly as errors. For a valid comparison, TROPOMI $XCH_4$ should be compared with total column measurements from ground-based instruments like TCCON. Please clarify whether the comparison shown is based on grid-to-grid analysis or a 50 km spatial average of TROPOMI data; this should be stated clearly in the main text or Figure 1 caption. Additionally, both datasets show a rising methane trend, which is encouraging and should be highlighted in the results.

   Regarding the 10% random perturbation applied, it is inappropriate to base this on differences between surface $CH_4$ and TROPOMI $XCH_4$ due to their different measurement scopes. Instead, perturbations simulating satellite uncertainty should rely on differences between TROPOMI and independent total column observations such as TCCON, to maintain physical consistency and justification.

4. While the manuscript addresses uncertainties related to TROPOMI observations, it does not account for uncertainties in wind speed, which are a critical component in top-down emission estimation. How does the use of boundary-layer averaged wind speed influence the emission estimates in your methodology? In the remote sensing community, it is standard practice to use multiple wind products and assess their associated uncertainties. However, I could not find any discussion or quantification of wind-related uncertainty in the current manuscript. Including this analysis would significantly strengthen the credibility and completeness of the results.

5. It is unclear why the robustness of the filtering method diminishes when the assumed TROPOMI uncertainty is increased to 20%. Could you clarify how these uncertainty threshold values were selected? Lines 260–264 are particularly difficult to follow and

should be revised for clarity. You mention that the R² value decreases under the 20% uncertainty assumption—please explain the underlying reason for this reduction. It would be helpful to elaborate on the relationship between the assumed uncertainty level and the resulting R² value.

6. In Figure 4, the bias after panel (e) is lower (0.56) compared to the filtered case (−1.62) when a ±10% perturbation is applied to TROPOMI XCH$_4$. However, the R² value remains largely unchanged between these cases. These metrics alone do not sufficiently demonstrate how the filtering or threshold choices influence the emission estimates. It would be helpful to provide additional analysis or metrics that better illustrate the impact of these filtering options on the accuracy and reliability of the emission estimations.

7. Section 2.4 is difficult to follow. Please consider splitting it into spatial filtering (removing grid cells with emission rates below 21.2 µg/m²/s) and temporal filtering. Also, clarify how the thresholds of 17, 25, and 31 µg/m²/s were chosen—are these related to specific percentiles? If Figure 3 illustrates the method, please reference it or include a schematic for clarity.

8. Have you considered using the median instead of the mean for the spatial filtering, since the mean can be influenced by regional transport and outliers? How does the emission estimate change when using the median?

9. Figure 6 d,e, f is missing.

10. Please provide the specific methane emission estimates reported by Cusworth (2021). How do your estimates compare to bottom-up inventories over the Permian Basin? Additionally, how do your results differ from other inversion-based CH$_4$ emission estimates in this region? There are also high-resolution satellite products like GHGSat and Carbon Mapper for facilities in the Permian Basin—how does your comparison with these datasets look? Do your estimates fall within their uncertainty ranges, or are there significant discrepancies?

11. The manuscript should include a clear quantification of the overall uncertainty associated with your method. Please provide an explicit uncertainty estimate and discuss its implications for your results.

12. Line 265 to 270: You mention that the largest positive values have been filtered out. Have you examined whether these locations correspond to known biases in surface albedo or aerosol optical depth (AOD)? High albedo or AOD can cause significant retrieval errors that might explain some of these extreme emission values. I strongly recommend incorporating supporting variables such as AOD, surface pressure, and albedo in your analysis. Without these contextual data, it is difficult to distinguish genuine emission signals from noise. Including these layers would enhance the transparency and robustness of your filtering method and better justify your data processing decisions.

Specific comments:
1. In Introduction, provide a detailed description of the inversion model used in this study and specify the geographical area where the inversion was applied. Instead of broadly stating that many studies use simple or complex models to estimate TROPOMI-based emissions, include specific examples of inversion models along with the regions they have been applied to, to better contextualize your approach.

2. Methodology: What type of trained model is used to calculate emissions? Are you referring to a machine learning approach or another type of model?

3. Line 127: You mention "such as the transport term and diffusion term in our equations and the divergence method used by others." Please specify who these "others" are by citing relevant studies or authors.

4. Line 131: The example given about XCH$_4$ values and uncertainties is confusing. With values of 1800 and 1900 ppb and 10% uncertainty, the ranges should be 1620–1980 ppb and 1710–2090 ppb respectively. Given this, how could the gradient become

negative or zero? Subtracting the lower and upper bounds still results in a positive gradient ranging from 90 to 110 ppb. Please clarify this point.

5. Line 154: What traditional technique are you referring to here? Please be specific.
6. Line 155: Is the perturbation applied consistently across all grid points or randomly across the chosen study domain?
7. Line 167: You state, "any positive value of the same magnitude or smaller is also due to uncertainty." Does this refer to the 21.2 µg/m²/s threshold or another value? Please clarify.
8. Line 177: What do you mean by "overlap between $CH_4$ and other species which are not resolved"? Which species are you referring to? Please specify.
9. Figure 4: The x-axis and y-axis labels are difficult to read—please increase their font size for better visibility. Additionally, the figure caption is confusing and does not clearly indicate which descriptions correspond to each subplot. I recommend revising the caption to be clearer and more concise, explicitly linking each part of the description to the respective subplots.
10. Line 259: Please separate the slope, $R^2$ value, and bias for clarity. Also, provide context explaining what Figures 4c and 4d specifically illustrate to help readers betterunderstand their significance.
11. Line 272: The statement about removing negative emission values or using the absolute value of the gradient is incorrect and needs revision. Studies like Maasakkers et al. (2021) and Shen et al. (2021) use inversion methods that do not apply the absolute value to the gradient. In fact, negative emissions commonly appear in normal inversion results but are retained in the analysis rather than removed.
12. Line 273: The phrase "same analysis approach" is vague. Please specify the exact method or approach you are referring to for clarity.
13. Line 37: Should it be "surface warming" or "global warming"? Please clarify.
14. Line 39: Use "gases" instead of "gasses."
15. Line 43: Provide the full names of TROPOMI, GOSAT, and SCIAMACHY before using the acronyms.
16. Line 44: Spell out the full forms of AGAGE, WMO, and TCCON.
17. Line 50: Use "extensive work to estimate uncertainty" instead of "extensive work on uncertainty."
18. Line 58: Specify what kind of in-situ processing and other processes are referred to; please explain in detail.
19. Line 62: Clarify what "significant non-linearity" means and provide quantitative uncertainty values for emission estimates.
20. Line 69: Define "clean area." Do you mean a background area without emissions?
21. Line 72: Specify the type of threshold and filter used, or refer to the section where this is detailed.
22. Lines 87–88: Clarify the phrase "The XCH4 retrieval used herein (version 2.4.0 Level 2) relies on a physical algorithm that factors in surface and atmospheric scattering." What exactly does this mean?

---

## Author Response (AR1)

Reviewer 1

The manuscript titled "How can we trust TROPOMI based Methane Emissions Estimation: Calculating Emissions over Unidentified Source Regions" presents an innovative perspective on computing emissions of methane using satellite data. The study proposes a new approach to calculate daily methane emissions using TROPOMI observations over areas without known sources that explicitly considering uncertainties in observed TROPOMI XCH4, and how these uncertainties propagate through the non-linear transformation that happens to extract emissions on a grid-by-grid basis.

Current methods for deriving CH4 emissions from satellite observations have relied on CH4 gradients, or complex chemical transport models. However, these methods ignore that uncertainties in observations of XCH4 can be transferred to the emission results through the CH4 spatial and temporal gradients. The work focuses on a region that is considered background in nature and has a long-term WMO observational site. The work adds white noise to represent the observed retrieval noise, and analyze the impact on emission results. The work then applied a filter to remove the white noise. They find that their emissions have no negative values, and very few large values, consistent with the effects of white noise in both directions. They also find that all of their emissions regions are consistent with known sources of emissions, while many grids otherwise computed to have emissions have no clear sources. They then apply this work to a known area with many methane sources in the USA to make a comparison with how the method functions in an area with many sources. Their results demonstrate that TROPOMI can be used to compute a lower minimum emissions threshold than previously thought using the standard calculation approaches based on plume identification and background subtraction.

I believe that their work is likely to be of substantial interest to the remote sensing, emissions, and methane mitigation research communities. I support the manuscript for eventual acceptance after addressing some specific comments below. The hope is that these comments will help the authors to more clearly explain their complexity and link more broadly with the larger community. I hope that this proposed moderate revision will enhance the approach's comprehensiveness and adaptability, and may help the paper revise to the level of an ACP highlight paper.

Specific Comments and Questions:

1. You claim that your method can identify emission sources that are consistent with the known facts on the ground and give some examples. Can you also make direct comparisons with existing bottom-up emissions inventories?

As shown in Figure RES-1, the values for WLG region shown in the EDGAR emissions inventory are very small, at least two orders of magnitude smaller than the results obtained. Furthermore, the EDGAR emissions are located in the wrong geospatial region by viewing satellite image maps, while all of our results are found on the bottom left or the bottom center. There are currently no emissions in EDGAR over any sources of our sources displayed in Figure 4. We have demonstrated using maps and in person observations that the regions identified are actually urban areas, villages, and so on. The simplest explanation is that the EDGAR emissions inventory can't identify smaller sources and miss.

What is important is that all the grid points where we identify emissions in this work can be explained scientifically, while many of the grid points identified as white noise in reality do not have any known anthropogenic or natural sources, and therefore are best explained as not being true emissions regions.

[Figure]

Figure RES-1: Annual mean methane emission fluxes for the Waliguan region in 2022 from EDGAR V8.0.

2. Please explain in more depth how you consider TROPOMI observational uncertainty overall. Do you partition the uncertainty into both the plume signal and the background separately, or consider them in tandem? How easily can the existing approaches based on background subtraction adapt your approach?

The following factors are among many but not all that influence TROPOMI observational uncertainty (noting that their influence varies as a function of location and time):

1. Overlap between CH4 and other species (the waveband resolution of the sensor),
2. Atmospheric conditions which reduce the incoming solar radiation in the wavebands relevant to CH4 (clouds and aerosols),
3. Atmospheric conditions which alter the band-by-band properties of incoming and scattered solar radiation in the wavebands relevant to CH4 (BC and CO).
4. Changes in the land-surface properties including but not limited to development, greening, industrial growth, agriculture changes, etc. (changes in surface reflectivity).

Computing emissions from concentration requires an additional step that includes knowledge of the wind, atmospheric transport and diffusion, in-situ processing and other processes. This set of net uncertainties may lead to changes in both the "background" conditions and the "plume" itself, which in net may cause the shape of the plume to change, or possibly even to completely disappear. Therefore, it is essential to compute how these uncertainties impact the real conditions when considering the spatial change in the observed fields. This includes not only the observed values themselves, but also whether or not the underlying assumptions of the model used are still relevant, or whether the simple idea of a plume model

may no longer be possible. For example, if the gradient switches direction when uncertainty is applied, the chances of the plume being real are significantly reduced, with the original detected plume more likely being just noise, rather than a real signal.

3.  This work claims to be capable of calculating a lower minimum emissions threshold than most other TROPOMI based CH4 emissions techniques currently published. However, this happens at the cost of detecting fewer overall plumes. You explain this by indicating that both large signals and small signals are excluded. While mathematically this seems reasonable, it is important to consider physical context. Can you add interpretation of what this may imply for other TROPOMI based methane emissions inversion studies, especially in other parts of the world?

This work demonstrates that the assumption of a "background enhancement" or a "stable plume" should be more carefully considered as the basis of mass balance computation. This work demonstrates that many times an "observed enhancement" may not be sufficiently robust to actually differentiate between the pixel(s) of interest and those surround it observational uncertainty is considered. This work addresses this point clearly by requiring that the spatial change in the observation must be sufficient to overcome observational uncertainty to actually be treated as a plume/source. Otherwise, is the plume anything other than white noise? It is extremely important to objectively and in a non-biased way investigate that signals detected are physical, not just a manifestation of observational uncertainty.

A second point this work demonstrates is consistent with the simple idea that any observed CH4 emissions which is negative (on a daily scale) is not physically realistic. Similarly, since uncertainty acts both in terms of positive and negative directions, any very high (or very low) computed emissions value should be treated with caution. Furthermore, any points which show up in a "plume" that are the same or higher than the base pixel (or much lower and not smooth) also need to be treated with caution. This work can address the issue of sources in adjacent or even two or three grids apart sometimes being in each other's downwind field, and sometimes not. However, current plume models do not address this point. Additionally, this approach can address the issue of uncertainties in both the wind and concentration gradients, allowing separation of these factors from observational noise. Similar aspects dealing with the shape and size plumes need to also be considered, and we believe that this work would help to address the uncertainties and issues with plume boundaries.

We believe that the approach outlined in this work will help present the scientific and policy communities with a next step towards clarifying the reliability of mass-balance computed emissions, attribution, and mitigation studies. We believe this topic is of vital importance to the scientific community, since present rapid mass balance approaches are being used to inform policy makers. Yet this work demonstrates a vast difference with the results of EDGAR. We hope that the findings herein demonstrate the added value that would be achieved if top-down emissions models in the future work to actively consider the effects of observational noise on the gradient term, and hence on the reliability of their computed emissions. We also hope that these findings demonstrate that the overall range of emissions computed may change, both that that the most extreme values (high and low) are more efficiently handled, thereby increasing

robustness.

4.  In the mass conservation equation used in the draft, the transport term is composed of the gradient of XCH4 multiplied by wind. The draft only considers the uncertainty in the XCH4 while not considering the uncertainty in the wind term. In line 281 of the draft, the author uses different thresholds to conduct sensitivity analysis. Putting these two together at different heights will lead to a more robust overall interpretation.

In this work, our main objective is to address the issue that when TROPOMI XCH4 contains a certain uncertainty, these uncertainties will propagate nonlinearly in the gradient term. We use the ERA5 wind field data with 0.25 resolution, which is widely used in the community. The resolution of TROPOMI XCH4 we used is 0.05º, so we interpolated the wind field data of ERA5 into the same grid as TROPOMI. In Equation 2, both the time gradient and the quadratic gradient are calculated using only XCH4, in the divergence term, the gradient calculation is by CH4 time wind, due to the re-interpolation of the wind field data, it means that the gradient calculation in the divergence term only calculates the gradient of XCH4, and the wind does not undergo gradient calculation. Moreover, this study mainly considers the impact of the uncertainty of XCH4 on the emission results. Regarding the use of wind field data, choosing the correct height of wind is very important. Previously, we only used the wind of one level (600hpa) to estimate emissions (WLG). We found that a single-level wind field cannot represent the wind field changes of the entire region. Now, our selection of wind field has been changed to choosing the corresponding height of the wind field based on the altitude of the grid.

1.  Sub-figure (b) is not described in the captions of Figure 1.

We have modified it, thank you for pointing it out

2.  On L17 the authors mention, "lower than existing community thresholds". Can the authors provide a specific number or range here?

500-8000kg/h/pixel (Dubey et al., 2023); 4200 kg/h/pixel (Jacob et al., 2016)

3.  On L87 please clarify which specific band is used by the TROPOMI XCH4 retrieval. Please highlight that different satellites use different bands.

TROPOMI has four spectral channels in the ultraviolet (UV), visible (VIS), near-infrared (NIR) and short-wave infrared (SWIR), with spectral ranges of 270–320,310–495, 675–775 and 2305–2385 nm, respectively. Using TROPOMI measurements in the NIR and SWIR for CH4 retrievals.

4.  On L162 please write more clearly about what values are filtered and at what level. This sentence is a bit confusing.

This work introduces a new two-step procedure to accomplish this task. First, we filter all grids in terms of their mean value, with any grids having values smaller than the 10th percentile of our data (in this case 21.2μg/m²/s) removed (We also used different thresholds for mean filtering to test the sensitivity of the model), to account for the fact that these grids likely are contributed to exclusively by white noise. Next,

the idea is extended to the temporal domain, specifically assuming that any negative values must be uncertain. Therefore, any positive value on that same grid which is of the same magnitude or smaller is also considered uncertain. Specifically, within each remaining grid, the most negative emission value computed is identified as $-\theta \mu g/m^2/s$, and all values from $[-\theta: \theta]$ $\mu g/m^2/s$ are excluded. We filter grid-by-grid, since the uncertainty is assumed to be an intrinsic property of each grid.

Reviewer 2

The authors present a novel methodology to calculate methane emissions with TROPOMI accounting for observational uncertainty. It is a very interesting topic and the idea of considering the uncertainty as part of the retrieval represents a significant contribution. However, the manuscript has important areas that need to be carefully reviewed. In general, they can be summarised as:

1) Methodology. The removal of pixels seems to apply a subjective threshold. That is, there is no uncertainty quantified and no consideration of spatial and temporal error correlation among pixels. Then, applying a baseline value is difficult to be applicable to any case.

We did not quantify the actual uncertainty of TROPOMI XCH4, which is beyond the capability of this work, what we did do is to create a framework of how to actively consider the effects of TROPOMI XCH4 uncertainty on emissions inversion. In order to compute an emission from a spatial dataset, both a temporal derivative and a spatial gradient operator are required. This work proposes a method by which to analyze how the column concentrations are nonlinearly propagated into emission results through the use of derivative and gradient terms, when satellite observation data contains certain uncertainties. This work proposes a new means to analyze the effects of this on the computed emissions, in an unbiased manner. This work is advancing the field into a new computational approach which moves beyond the idea of background subtraction. This seeks to overcome a critical issue in that background subtraction inherently does not consider: the observational uncertainty of the background, temporal variation, and spatial variation terms.

The basis of the work is to first compute all possible emissions values using the mass balance equation, on a day-by-day and grid-by-grid basis. We then develop a method by which to separate those emissions which are actual and robust and real when considering the observational uncertainty, from those which are computed as emissions but in fact are merely computed as a result of the noise within the observed signal. Since negative emissions are not physically realistic or possible (given the loss terms, OH, Cl radicals, and bacteria operate on time scales of a decade or more, and this work computes emissions day-by-day), any negative value must be due to noise. However, merely removing negative values is not mathematical, since white noise is a random uncertainty applied to the observed data, and is equally likely to be positive or negative. Therefore, any values which fall in the probability distribution which is best fit by a normal distribution with a center at zero, are associated with this noise. They are computed emissions values which are mathematically valid, but are not physically realistic, instead they are purely computed as a result of the noise associated with the observational uncertainty.

In reality, the observed distribution looks mostly normal, matching the distribution of white noise, which has a normal distribution with an approximate average of 0 and a range around zero which is roughly equal in both positive and negative space. But upon careful examination it was found to be slightly positively biased. The cutoff is applied based on where the PDF diverges from a normal shape on the positive side, which is consistent with a first order approximation of where the true non-noise signal should be found. Similarly, after this first filtration has been done, a second filtration is applied with respect to time on all of these pixels. If all computed values are positive, they are retained. If any are negative, then all values between this negative value and the same absolute value are likely noise based on the same argument. All unphysical values and all values computed merely due to observational noise are henceforth removed from further consideration.

This new approach is of vital importance to the scientific community, since at the present time such rapid mass balance approaches are being used to inform policy makers, yet this work demonstrates that a vast majority of the computed emissions being reported and published are actually just noise due to the observational uncertainty and the non-linearity of the gradient term used to compute emissions.

2)  Validation. There is a temporal and spatial mismatch between the reference data and the retrieved results. In consequence, the comparison presents significant differences (more than double in many cases) and results in an inconclusive validation.

When we responded to the comments raised by the reviewers, we found that the north-south direction of the emission plots in our preprint was reversed. We apologize for this. In the revised version, we have modified it and redrawn it.

Considering the significant terrain changes in this area, we instead select the corresponding wind at different altitude levels based on the altitude, with a height range of 950hpa to 750hpa, and remove the abnormal wind with a speed greater than 10m/s, the calculated emission is shown in Figure 6a, negative emissions have occurred in some areas. Previous studies also showed negative emissions when estimating methane emissions in the Permian basin (Schneising et al., 2020; Veefkind et al., 2023), Veefkind et al., (2023) employed background subtraction to filter out negative emissions and smaller emission signals in Permian basin and believed that the intense variations in topography and surface albedo may lead to significant negative emissions. However, these studies did not further analyze the causes of negative emissions. Simple background filtering methods cannot effectively remove all emission noise.

Due to the seasonal variations in emissions in this area (Varon et al., 2025), the original time filtering method was overly strict. After performing spatial mean filtering, we changed the time to monthly filtering. The emission results after filtering are shown in Figure 6c. Figures 6b and d are the PDFS corresponding to Figure 6a and c respectively. Due to the intermittent nature of facility emissions and the inclusion of multiple emission sources in one emission grid, our overall emission results are larger than those of the dataset by Cusworth et al. (2021). However, some of our results still match the dataset well. The grey bars indicate the emission error range of these points. Below 2500kg/h/grid emissions,

there is a good match between the two. Above 4000kg/h/grid emissions, there is a significant deviation between the two. However, the emission values of Cusworth are all found within the range of daily emissions calculated on the same grid using our methodology, meaning that differences are possibly due to temporal representation.

[Figure]

Figure 6: (a) Annual mean methane emission fluxes from the Permian Basin for 2019-2020, (b) PDF of the Permian Basin emission fluxes for 2019-2020, (c) Annual mean methane emission fluxes from the Permian Basin after filtration in 2019-2020, (d) PDF of the 2019-2020 methane emission fluxes from the Permian Basin after filtration. Backgrounds of (a) and (c) are from Esri World Imagery.

[Figure]

Figure 7: (a) Annual mean CH4 emissions (kg/h/TROPOMI grid) on grids which overlap spatially with an emission observed by observed by Cusworth (2021); (b) The red dots in the scatter plot represent the

points where our emission results overlap with those of Cusworth (2021) in both space and time and the difference is within 2 times. The black dots represent our emission results that overlap in space but do not overlap in time. The gray bars indicate the error range. Backgrounds of (a) is from Esri World Imagery.

3) Figures. Figures 6D, E, F are mentioned but they cannot be found in the manuscript. The preprint has been downloaded again but could not find them.
Thank you for your point. We have already made supplements

Specific comments are here included:

- L92, is the resampling affecting the data and uncertainty? given that emission might represent a small cluster of pixels, it might result in artifacts (depending on the resampling) and impacts the uncertainty.

  Thanks for your suggestion, we have noticed that resampling does indeed have an impact on data and create uncertainty, although most current studies still use resampling to process data (Varon et al., 2023; Liu et al., 2021). we are preparing to use swath-by-swath data in future work. The resolution after resampling we used is 0.05 degrees, which is close to the resolution of $5.5*7km^2$ of TROPOMI XCH4. This means that the values of the resampled mesh are mainly affected by its adjacent original mesh. This change is limited within the range of numerical differences between adjacent meshes and will not generate new values that exceed the extreme values of the original data. These errors are all within the 10% uncertainty range of our application. This is all the more rationale to support that the 10% uncertainty is a reasonable figure to employ, and supports the methodology used herein. We do understand that resampling across different days or across multiple grids in length will introduce additional errors and have not done either practice specifically for this reason, although both are commonly done by the emissions inversion modeling community.

- L95- 97 The webpages for the products are included at the end of the manuscript. However, many of them could be directly referenced with a DOI.

  We have modified these references. Thank you for pointing it out. In actuality, these websites are from the EU and the USA, which until recently were considered reliable. We have instead switched to DOI so that this will not remain an issue, and that we follow best practices.

- L100-106 It would be desirable to understand the uncertainty associated to the ground truth. Is it negligible compared to the satellite data?

  We have used a universal open path gas analyzer (LI-7700) with an uncertainty of about 5ppb to analyze the temporal and spatial variation of concentration and computed emission of CH4 data at high temporal frequency in Shanxi Province (Lu, F et al., 2025). These variation in the concentration data and resulting emissions observed within a 1x1 $km^2$ to 5x5$km^2$ area is very large, indicating that satellites may suffer from such issues. However, when addressing the

results on a day-to-day basis, we find that we can still constrain the uncertainty to within 30-50%. We also have computed emissions of CH4 using both surface and satellite data in other studies, and again found that uncertainty in final emissions products tends to nearly always be larger than 10% (Hu et al., 2024).

Finally, remember that before any gas products can be inverted, that aerosols and clouds must first be considered. Even in cases where their retrieval is successful, their uncertainty is at least 5% (Torres et al., 2020; Tiwari et al., 2023; Tiwari, et al., 2025). Therefore, any uncertainties in CH4 retrieval must in fact be larger than this, placing a minimum uncertainty bound on observations of 5%, again demonstrating that our choice is reasonable.

- L119 "field" instead of "filed"

   We have modified it, thank you for pointing it out.

- L131 the uncertainty of 10% should be fully uncorrelated spatially. defining a 10% uncertainty with no consideration of spatial error correlation is limited.
   Yes, we agree. All of our uncertainties are applied fully uncorrelated spatially in the present version of the manuscript. We give each grid a random error within the range of 10%, we cannot find where we have said differently. Please let us know specifically where such a statement was made otherwise, and we will fix it.

- L168 The criteria to define a threshold is based on the lowest sampled. This could be very unstable to expand to other cases.

   This is a very important point/issue. Since the error is based on the individual retrieved pixel observation on each day and time, therefore the uncertainty must be based on each such value. However, when computing emissions, the computation requires more information than just that local pixel in space and time. It requires information about the previous or next local pixel information (can be plus or minus 1 day – (Lu et al., 2025; Li et al., 2023; Hu et al., 2024; Qin et al., 2023) or can even just be plus or minus 100minutes (He et al., 2025), as well as adjacent pixels' information from the same day and time. It also includes information about the wind field and its gradient. For this reason, the statistics of any computed emissions must be considered locally. It is not reasonable to use a global number. However, this paper outlines a mathematical and analytical approach which can be applied elsewhere.
   We use a threshold to filter the mean of the emission grid, which is based on the point at which the shape of the computed emissions distribution diverges from a white-noise induced normal distribution centered at zero. Please see the details above. The main purpose of the first step of filtering is to separate the grids with emission signals which are genuine on average, from those pixels which on average behave as white noise. The main purpose of the second filtering step is to separate the temporal subset of white noise from the remaining grids after the first filtering step was applied. Please see details above. The threshold used in the second step of filtering is dynamic and can achieve unbiased filtering.

This methodology is then applied in two vastly different areas, and demonstrated to yield reasonable results in both areas. If you are interested, please pick up this approach and apply it to your area of choice, and then help build on the current results/findings.

- L183-195 Which is the conclusion about AOD uncertainty? It depends on AOD level and type among other things. Which is the implication in your uncertainty budget?

I do not know what you mean by AOD type? I do know that AOD is defined by wavelength, and is the exponential loss of the solar radiation through scattering and absorption between the top of the atmosphere and the surface.
We have published some works on this topic, and believe that both AOD and AAOD contribute to the uncertainty. We know from Torres et al. (2020) that the AOD uncertainty is a minimum of 5%. We also believe that this uncertainty is a function of the underlying aerosol microphysical properties (i.e., size, composition, and mixing state) as well as column properties (column number loading, mass loading, etc.) (Wang et al., 2021; Cohen and Wang, 2014; Tiwari et al., 2023).
We have selected a number of 10% since it is quite high compared to what the CH4 retrieval community currently uses, but is quite low compared with most other TROPOMI products(NO2 from 10% to 40% (Boersma et al., 2018; Pollard et al., 2022), black carbon aerosol (BC) from 20%-50% (Vignati et al., 2010; Tiwari et al.,2023, 2025), and CO starting from a minimum of 10% to 20% and upwards). We have also performed the same runs with a 20% uncertainty, as described in Figure 4.

- L205 and Fig4 There is a systematic bias between the two of them of approx 100ppb (that is correctable) and about 30ppb random component. However, the uncertainty is defined by the 100ppb. Could be this corrected and assume the 30ppb random component to verify the results? Which are the implications of a temporal correlated error in the dataset?

We are very confused by the question. Line 205 talks about observations of CH4 from Waliguan and from TROPOMI. Figure 4(Corresponding to Figure 5 in the revised draft) talks about computed emissions of CH4. These are apples and oranges and cannot be compared.
With regards to bias correction of observed values (i.e., L205) we have not performed any bias correction to any concentration values. We use the given products and values, since this is what the community has access to and uses. Since these are not our own measurements, and these have been extensively published by other groups including the WMO, we expect the data validation and quality to already be reasonable. We just use this to demonstrate that there is in fact an uncertainty in the observations.
In terms of Figure 4 (Corresponding to Figure 5 in the revised draft), these are computed emissions values, as explained above. Negative emissions values are not physically possible. Therefore, bias correcting them does not have any physical basis.

- Figure 3. The same scale for both plots would be helpful.

We have modified it, thank you for pointing it out.

- L329 Seems that Fig6B and 6C have been swapped in the text.

  Thank you for your point. We have already made supplements.

- Fig 6C The authors used a log axis and this does not clearly shows the large differences between the reference and retrieved emissions. A linear axis would be desirable.

  Thanks for your suggestion. We have already modified it.

Reviewer 3

How can we trust TROPOMI based Methane Emissions Estimation: Calculating Emissions over Unidentified Source Regions

The author presents a methodology for estimating methane emissions using TROPOMI data while accounting for observational uncertainty. These uncertainties are crucial, as they can significantly influence emission estimates. However, there are several critical aspects of the methodology that need to be carefully reviewed.

1. Selecting pixels with QA > 0.5 may introduce artefacts, potentially increasing uncertainty in the emission estimates. Rather than relying solely on this threshold, did you explore the use of AOD and albedo filters as recommended by Schuit et al. (2023) and Nesser et al. (2024)? If so, how did these alternative filtering approaches impact your results? I strongly recommend using only pixels with QA = 1.0, as lower QA values may include retrieval artefacts. Additionally, did you consider using the blended TROPOMI+GOSAT product, which reduces biases through the integration of GOSAT data? This could further enhance the robustness of your emission estimates.

   First, our group has published multiple papers regarding aerosols, and carefully accounts for these effects (i.e., Wang et al., 2021; Tiwari et al., 2023, 2025; Liu et al., 2024; Liu et al., 2024). We have checked the MISR AOD and AAOD values to ensure that there was not a substantial amount of interference on the days or times, or we have otherwise filtered the data. Second, our idea was to use common products that the community are currently using, which have claimed precisions which are very high, and then actively considering the uncertainties. This is a philosophical choice, since while there will always be new and smarter ways to analyze data and reduce its uncertainty, there will always be some uncertainty that remains. Furthermore, we wanted to focus on common uncertainty levels today, since these are the products that the vast majority of emissions estimates have relied upon over the past few years as published in high end journals (Shen et al., 2023; Hemati et al., 2024). We also are aware that at the present time there are multiple versions of TROPOMI CH4 products (Sicsik-Paré et al., 2025). However, we are also happy to cite these papers to demonstrate that the CH4 inversion community is continually improving.

   We do agree that using higher quality control is better. However, there are two issues as to why we did not use QA=1. First, there is insufficient data to compute emissions at any of our locations. As observed, after filtering for the observational noise from the computed emissions

signals, our existing products already contain very few reliable emission inversions. Secondly, the community as a whole is tending not to use such a high QA value. Again, our concept was philosophically to address the data that is currently being used by the community.

Thank you as well for your comment. We actually use GOSAT for other work. At the present time, we have not considered using GOSAT in this work, as the change in the amount of data would be small, and it would introduce a second product with a likely different uncertainty range. Furthermore, there is no data from GOSAT which overlaps with our surface WMO station, and therefore we could not find a way to look at its uncertainty with respect to our surface observations.

2. Figure 2: Typically, pixels classified as water surfaces—especially those near coastal areas— are excluded due to their high uncertainty and potential for retrieval errors. Including TROPOMI pixels over water bodies can lead to unreliable or biased emission estimates, as these pixels often suffer from issues related to surface reflectivity and retrieval sensitivity. Additionally, the manuscript does not show a 2D map of TROPOMI $XCH_4$ in the Waliguan region and Permian Basin, which would help in evaluating the spatial context and data quality. It would also be beneficial to include maps of supporting variables such as albedo, AOD and surface pressure in this region to assess the robustness of the retrieval and filtering process. I recommend adding these visualizations and clarifying how water-body pixels were handled in your analysis. Can you also add the TROPOMI observational density maps?

When we responded to the comments raised by the reviewers, we found that the north-south direction of the emission plots in our preprint was reversed. We apologize for this. In the revised version, we have modified it and redrawn it.

Our final emissions results do not contain any emissions on or over water bodies. Therefore, we believe that our approach has successfully accounted for the potential errors that you have discussed, and done so in a more physically consistent manner.

We have now included average maps of XCH4, TROPOMI observational density, height from DEM and AAOD (440nm), AAOD (880nm), AAOD (440nm)/AAOD (880nm) and add them to the supplement. As outlined in our papers above, we believe that AAOD (440nm) and AAOD (880nm) are sufficient to explain large potential errors in XCH4.

The distribution of the emission grid in Figure 4a is consistent with the area with a high TROPOMI observation density (Figure S1) in this region. Negative emissions are mainly concentrated in areas A and C of Figure 4a. By combining the AAOD_440nm, AAOD_880nm of this area and the ratio between them (Figure S1), we find that the area with the highest negative value in area A corresponds to the high-value area of AAOD_440nm/AAOD_880nm. Areas with a high ratio indicate that there is a considerable amount of sand and dust in that region, this will have an impact on the inversion of TROPOMI. Through comparison with the actual situation, it was found that this area is indeed a desert/wasteland area. Area C is mainly composed of a large desert (Gemutan) and the surrounding villages and towns. Negative

emissions and smaller emission signals (white noise) are located in the desert and at its edge.

[Figure]

Figure S1. (a) Annual average methane column concentration of Waliguan region; (b)TROPOMI XCH4 observational density in Waliguan region; (c) Surface pressure in the Waliguan region; (d) The average AAOD_440nm of Waliguan region; (e) The average AAOD_880nm of Waliguan region; (f) The ratio of AAOD_440nm to AAOD_880nm.

[Figure]

Figure S2. (a) Annual average methane column concentration of Permian basin; (b)TROPOMI XCH4 observational density in Permian basin; (c) Surface pressure in the Waliguan region; (d) The average AAOD_440nm of Permian basin; (e) The average AAOD_880nm of Permian basin; (f) The ratio of AAOD_440nm to AAOD_880nm.

3. Lines 203–205: TROPOMI measures the total column-averaged dry-air mole fraction of methane ($XCH_4$), whereas the Waliguan station provides surface-level $CH_4$ These are fundamentally different metrics, so differences between them should not be interpreted directly as errors. For a valid comparison, TROPOMI $XCH_4$ should be compared with total column measurements from ground-based instruments like TCCON. Please clarify whether the comparison shown is based on grid-to-grid analysis or a 50 km spatial average of TROPOMI data; this should be stated clearly in the main text or Figure 1 caption. Additionally, both datasets show a rising methane trend, which is encouraging and should be highlighted in the

results. Regarding the 10% random perturbation applied, it is inappropriate to base this on differences between surface CH₄ and TROPOMI XCH₄ due to their different measurement scopes. Instead, perturbations simulating satellite uncertainty should rely on differences between TROPOMI and independent total column observations such as TCCON, to maintain physical consistency and justification.

To ensure the quality of the data, TCCON stations are always located in areas with better observation conditions. Although the data from these TCCON stations have a good match with TROPOMI, they do not represent all areas, especially those with poor observation conditions such as high altitudes and high aerosols.

Given the height, atmospheric conditions, pressure, and other surrounding geography at Waliguan, all of the observations are within the free troposphere (WMO-Background Station). This is the reason why this site was selected by the WMO as a global background station. If there is a TROPOMI retrieval error of assuming that the surface is different from the free troposphere which the TROPOMI team applied, then this is a type of important error that we address within our parameterizations. Again, the values at this height are all observations directly of the free troposphere, and hence are physically equivalent to global averaged tropospheric column XCH4.

Finally, remember that before any gas products can be inverted, that aerosols and clouds must first be considered. Even in cases where their retrieval is successful, their uncertainty is at least 5% (Torres et al., 2020; Tiwari et al., 2023; Tiwari, et al., 2025). Therefore, any uncertainties in CH4 retrieval must in fact be larger than this, placing a minimum uncertainty bound on observations of 5%, again demonstrating that our choice is reasonable.

We have done considerable ground/column/XCH4 analysis in the past. We do not think that a 50km or other spatial average is reasonable, and we never do this with CH4. We always use a grid-by-grid analysis and comparison. We already have published work about how high frequency surface observations at the scale of 1-3km can yield different signals even within a single TROPOMI grid in high emissions areas (Lu, F et al., 2025). The only times we have considered spatial extension is under cases where we have strong physical grounds (i.e., Tiwari et al., 2025), and we still do not do averaging.

1. While the manuscript addresses uncertainties related to TROPOMI observations, it does not account for uncertainties in wind speed, which are a critical component in top-down emission estimation. How does the use of boundary-layer averaged wind speed influence the emission estimates in your methodology? In the remote sensing community, it is standard practice to use multiple wind products and assess their associated uncertainties. However, I could not find any discussion or quantification of wind-related uncertainty in the current manuscript. Including this analysis would significantly strengthen the credibility and completeness of the results.

Our approach is a model free analysis which actively calculates first order wind-based advection and pressure-based advection, as well as first order diffusion, similar to WRF, GEOS, and other regional and global scale models. The main purpose of this paper is to study the impact of the uncertainty of satellite data on emission results, the uncertainty of the wind field will be taken into account in subsequent research. We do not use wind linearly, in Eq (2), we compute the time gradient of XCH4, the gradient of the wind times the concentration and the quadratic gradient of XCH4. Our calculation is based on the gradient of XCH4, not that of the wind, which means that the generation of white noise is merely due to the nonlinear propagation of the XCH4 error in the gradient term, which is what we describe throughout the paper. Due to the significant terrain changes in this area, previously we used the wind at a single altitude level, which might have been insufficiently representative. Now, we have changed to selecting the wind at the corresponding altitude level based on the altitude of each grid. However, we have already demonstrated in previous work for NOx, CO, BC and CH4 emissions in polluted regions (Qin et al., 2023; Lu et al., 2025; Lu et al., 2025 [preprint]; Li et al., 2023; Li et al., 2025; Liu et al., 2023; Hu et al., 2024) that while this issue is relevant, that it is less impactful than using the gradient term correctly, and in the case of NOx, the issue of retrieval error. This paper is the first attempt to bring the issue of retrieval error into the retrieval of CH4 emissions. We did not use a boundary-layer wind at all.

2. It is unclear why the robustness of the filtering method diminishes when the assumed TROPOMI uncertainty is increased to 20%. Could you clarify how these uncertainty threshold values were selected? Lines 260–264 are particularly difficult to follow and should be revised for clarity. You mention that the $R^2$ value decreases under the 20% uncertainty assumption—please explain the underlying reason for this reduction. It would be helpful to elaborate on the relationship between the assumed uncertainty level and the resulting $R^2$ value.

Our filtration method is to filter the emissions estimated using the mass conservation equation. When we randomly disturb each grid, it means that each grid all contains a certain degree of uncertainty. As the degree of disturbance increases, the difference between the estimated emissions and the undisturbed emissions will also increase, but as the number of perturbations increases, these differences will decrease. Due to the large perturbation estimation time, we only perturbated 1,000 times here. This is also the reason why we conduct perturbations, after multiple perturbations, some uncertainty in the satellite data will be smoothed out.

We filter out some physically unreasonable white noise, but because the degree of disturbance is relatively large, the filtered emission results have certain differences from the undisturbed emissions. As uncertainty increases, the $R^2$ will decline, but the decline is not significant. The main purpose of our filtration method is to obtain as accurate and reliable emission results as possible and remove white noise caused by data errors. It should be emphasized that we did not attempt to reduce the uncertainty of data inversion, but rather to minimize the impact of data uncertainty on the emission results.

3. In Figure 4, the bias after panel (e) is lower (0.56) compared to the filtered case (−1.62) when a ±10% perturbation is applied to TROPOMI XCH4. However, the $R^2$ value remains largely unchanged between these cases. These metrics alone do not sufficiently demonstrate how the

filtering or threshold choices influence the emission estimates. It would be helpful to provide additional analysis or metrics that better illustrate the impact of these filtering options on the accuracy and reliability of the emission estimations.

In Figures 5a, b, e and f, due to the presence of a large number of emission values close to 0, $R^2$ doesn't change significantly. Our filtered emission (Figure 5c) is 70% less in number than the unfiltered emission (figure 5a) and 51% less than the emission after removing negative values. After removing a large number of values close to 0, as well as some negative emissions and a few maxima, the R of our emission results remains stable, and the RMSE has only increased by 0.3. This is sufficient to demonstrate the robustness of our filtering method. Many studies have mentioned that negative emissions are non-physical emissions and must be removed, but they have not further analyzed the causes of negative emissions. The methods to remove negative emissions are merely background filtering by setting a certain threshold (we have updated the specific research methods in Introduction). Our filtration method fully takes into account the causes of these emission noises and removes them through unbiased filtration.

In the first step of filtering, we now select the 65th percentile of the emission value(26.5ug/m²/s) (considering that the distribution of emission values is positively skewed, Figure 2) as the threshold for spatial filtering. We also used the 50th percentile emission value (12.4 ug/m²/s) as the threshold for filtering (Figure S4). We found that some emissions occurred in unreasonable areas (deserts), which also indicates the necessity of threshold filtering in space and the importance of how to choose the threshold. Choosing a threshold that is too low will cause the emission results to include some grids composed of emission noise. Choosing an excessively high threshold will result in the omission of some emission sources

4. Section 2.4 is difficult to follow. Please consider splitting it into spatial filtering (removing grid cells with emission rates below 21.2 μg/m²/s) and temporal filtering. Also, clarify how the thresholds of 17, 25, and 31 μg/m²/s were chosen—are these related to specific percentiles? If Figure 3 illustrates the method, please reference it or include a schematic for clarity.

Section 2.4 mainly describes our two-step filtering method. Thanks for your suggestion,I agree to split it into spatial filtering and temporal filtering, we have modified it in manuscript.

The emissions estimated using the mass conservation equation have a value of -31μg/m²/s at the 10th percentile, -25μg/m²/s at the 15th percentile, -21μg/m²/s at the 20th percentile, and -17μg/m²/s at the 25th percentile. Considering that white noise follows a normal distribution, we select the absolute values of these values as thresholds of spatial filtering and conduct sensitivity tests. After that, we changed the way we selected the wind field and re-estimated the emissions. Above, we have already explained in detail the threshold we are now using and how to make the selection. The emission distribution of spatial filtering using different thresholds is presented in the supplement.

5. Have you considered using the median instead of the mean for the spatial filtering, since the mean can be influenced by regional transport and outliers? How does the emission estimate change when using the median?

According to your suggestion, we used the median instead of the mean for spatial filtering. The result is shown in Figure RES-2. Figure RES-2 is the median of each grid of emissions, which has a slight difference from the emission mean. Figure RES-2b is the filtered emission. Compared with spatial filtering using the mean, the overall difference between the two is not significant. However, some emission grids emerged in the desert area of region C using median filtering. We examined these grids and found that the number of emissions in time was less than that of other grids, which might be due to regional transport. Overall, it might be more appropriate to choose the mean as the filtering criterion.

[Figure]

Figure RES-2: (a): Median of emission results; (b) The emission using the median of the emission results instead of the mean for spatial filtering and then perform temporal filtering.

6. Figure 6 d,e, f is missing.

   Thank you for your point. We have already made supplements and added this in supplements.

7. Please provide the specific methane emission estimates reported by Cusworth (2021). How do your estimates compare to bottom-up inventories over the Permian Basin? Additionally, how do your results differ from other inversion-based $CH_4$ emission estimates in this region? There are also high-resolution satellite products like GHGSat and Carbon Mapper for facilities in the Permian Basin—how does your comparison with these datasets look? Do your estimates fall within their uncertainty ranges, or are there significant discrepancies?

   Thanks for your suggestion. We have not compared them with the existing bottom-up inventories, which have significant errors and miss many medium or small emission sources. We have added the Cusworth (2021) emission estimates to the supporting documents. We have redrawn the scatter plot comparing the emissions with Cusworth (2021). This includes the range of uncertainties. When conducting spatio-temporal matching, Cusworth (2021) has a good match with our emission results and is within our emission range.

8. The manuscript should include a clear quantification of the overall uncertainty associated with your method. Please provide an explicit uncertainty estimate and discuss its implications for your results.

Since we cannot determine exactly how much uncertainty TROPOMI contains in some regions, in this work, we only roughly estimate the approximate range of uncertainty in TROPOMI data based on other species such as BC, CO, etc., as well as comparisons with s ground station observation data (WMO). In this paper, random perturbations with 10% and 20% uncertainty are used. Our approach mainly filters the impact of the uncertainties of these data on the emission results. The uncertainty range of our emission estimation method has been plotted in the emission scatter plot (Figure 7b) compared with Cusworth (2021). When training our model, we obtained a total of 924 sets of coefficients, which means that at each point, we can estimate 924 emission values. We set the uncertainty range to the 20th to 80th percentile values of these 924 emission values as the uncertainty range of the emission values at this point.

9. Line 265 to 270: You mention that the largest positive values have been filtered out. Have you examined whether these locations correspond to known biases in surface albedo or aerosol optical depth (AOD)? High albedo or AOD can cause significant retrieval errors that might explain some of these extreme emission values. I strongly recommend incorporating supporting variables such as AOD, surface pressure, and albedo in your analysis. Without these contextual data, it is difficult to distinguish genuine emission signals from noise. Including these layers would enhance the transparency and robustness of your filtering method and better justify your data processing decisions.

Maximum values are filtered out in two cases: 1. The emission value of the grid is only a few days, there are several maximum emission values, and at the same time there are several maximum negative emission values. The mean of this grid is relatively low and it is removed in the first step of spatial filtering. 2. The grid has a maximum value, and at the same time, there are negative emissions with an absolute value larger than this maximum value, which are removed in the second step of time filtering. Thanks for your suggestion. We have added the AAOD of the study area and conducted the analysis above.

Specific comments:

1. In Introduction, provide a detailed description of the inversion model used in this study and specify the geographical area where the inversion was applied. Instead of broadly stating that many studies use simple or complex models to estimate TROPOMI-based emissions, include specific examples of inversion models along with the regions they have been applied to, to better contextualize your approach.

Thanks for your suggestion, we have made modifications in Introduction. The modifications are as follows:

Computing emissions from concentration requires an additional step that includes knowledge of the wind, atmospheric transport and diffusion, in-situ processing and other processes (Cohen and Prinn 2011; Cohen et al., 2011). There are many studies which have used various complex models such as GEO-Chem (Hancock et al., 2025; Nesser et al., 2024; Varon et al., 2023) and simple models such as the Gaussian integral method (Schneising et al., 2020), Divergence method (Liu et al., 2021; Veefkind et al., 2023), mass balance method (Hu et al., 2024) to attempt

to estimate methane emissions from TROPOMI in near real time. However, due to the uncertainties in the retrieved TROPOMI data and wind fields, the computation of gradient term(s) necessary to compute emissions may lead to significant non-linearity (Cohen and Prinn 2011; Cohen et al., 2011; He et al., 2024), thereby generating negative emissions and smaller emission values. Hancock et al. (2025) prevented non-physical negative emissions by inverting prior emissions using the lognormal error probability density function (PDFS), Schneising et al., (2020) defined the background from a 2°× 4° headwind box and determined negative emissions or very low emissions for many days in the Permian basin, despite strict filtering criteria for background observations. Veefkind et al., (2023) employed background subtraction to filter out negative emissions and smaller emission signals in Permian basin and believed that the intense variations in topography and surface albedo may lead to significant negative emissions. However, these studies did not further analyze the causes of negative emissions. Simple background filtering methods cannot effectively remove all emission noise. Therefore, finding ways to address how observed and modeled uncertainties lead to the robustness of inverted methane emissions is a necessary and essential step to gain trust in resulting emission quantification and usefulness for attribution (Li et al., 2025; Tiwari et al., 2025). To gain confidence, such emissions should include methods which are both unbiased and robust, and are capable of identifying sources both known and unknown, monitoring emissions from those sources, and validate whether emissions are actually being reduced or eliminated (Hemati et al., 2024).

2. Methodology: What type of trained model is used to calculate emissions? Are you referring to a machine learning approach or another type of model?

   We conducted spatio-temporal matching of the ground observation data with the time gradient term, transport term and diffusion term calculated using TROPOMI XCH4 data, then extracted them using least squares method of multiple linear regression.

3. Line 127: You mention "such as the transport term and diffusion term in our equations and the divergence method used by others." Please specify who these "others" are by citing relevant studies or authors.

   We have added it, thank you for pointing it out

4. Line 131: The example given about XCH₄ values and uncertainties is confusing. With values of 1800 and 1900 ppb and 10% uncertainty, the ranges should be 1620–1980 ppb and 1710–2090 ppb respectively. Given this, how could the gradient become negative or zero? Subtracting the lower and upper bounds still results in a positive gradient ranging from 90 to 110 ppb. Please clarify this point.

   To explain more clearly, here we have drawn an example graph(Figure RES-3). The red grid represents the grid with emission sources, the yellow grid in the middle is its plume, and the

blue grid is the background grid. 1800ppb may become any number within the range of 1620 to 1980ppb and 1900ppb may also become any number within the range of 1710 to 2090ppb when there is 10% uncertainty. The original 1800 and 1900ppb may become 1850 and 1900ppb, the direction of the gradient changes, but the wind direction remains the same, so the calculated emission result is negative. In the background concentration area of methane without emissions, due to the uncertainty of the data, some emission noise with emission values close to zero is caused.

[Figure]

Figure RES-3. An example of nonlinear propagation of data error in the gradient term

5. Line 154: What traditional technique are you referring to here? Please be specific.

   Thank you for pointing it out, we have modified it.

6. Line 155: Is the perturbation applied consistently across all grid points or randomly across the chosen study domain?

   We added a random perturbation within the range of 10% and 20% to all the grids in the study area, and the perturbation number is randomly across the study domain.

7. Line 167: You state, "any positive value of the same magnitude or smaller is also due to uncertainty." Does this refer to the 21.2 μg/m²/s threshold or another value? Please clarify.

   It does not refer to the threshold of 21.2ug/m²/s. First, we use a threshold to filter the mean of the emission grid. Due to the nonlinear propagation of uncertainty in the gradient term, the resulting emission noise follows a normal distribution, in the absence of emission sources, the mean emission of this grid should be close to zero. Therefore, we use a threshold to screen the grids spatially and remove those without emission signals, these estimated emission values of these grids are all noise. The main purpose of the first step of filtering is to separate the grids with emission signals from those without.

The main purpose of the second filtering step is to separate the emission noise from the emission signal in the remaining grid after the first filtering step. When a grid has negative emission values over time, considering that the emission noise follows a normal distribution, the values between the minimum and the absolute value within this grid should all be classified as emission noise, so any positive value of the same magnitude or smaller in the range is also due to uncertainty.

In addition, we have made modifications to this part, detailing how we select the threshold

8. Line 177: What do you mean by "overlap between $CH_4$ and other species which are not resolved"? Which species are you referring to? Please specify.

Here it refers to CO. The bands used for the inversion of CO and CH4 have overlapping parts

9. Figure 4: The x-axis and y-axis labels are difficult to read—please increase their font size for better visibility. Additionally, the figure caption is confusing and does not clearly indicate which descriptions correspond to each subplot. I recommend revising the caption to be clearer and more concise, explicitly linking each part of the description to the respective subplots.

Thank you for your suggestion. We have already made some revisions to it.

10. Line 259: Please separate the slope, $R^2$ value, and bias for clarity. Also, provide context explaining what Figures 4c and 4d specifically illustrate to help readers betterunderstand their significance.

Thank you for your suggestion. We have already made some revisions to it.

11. Line 272: The statement about removing negative emission values or using the absolute value of the gradient is incorrect and needs revision. Studies like Maasakkers et al. (2021) and Shen et al. (2021) use inversion methods that do not apply the absolute value to the gradient. In fact, negative emissions commonly appear in normal inversion results but are retained in the analysis rather than removed.

Thank you for pointing it out, we have modified it

12. Line 273: The phrase "same analysis approach" is vague. Please specify the exact method or approach you are referring to for clarity.

The same analysis approach is referring to setting a certain minimum detection threshold to filter the emission results or only retain positive values. Setting a threshold to remove emission values below this threshold will result in the elimination of some smaller emission signals, while only removing negative emissions will retain a large amount of smaller emission noise. Here, the same analysis approach is to only retain positive emission values.

13. Line 37: Should it be "surface warming" or "global warming"? Please clarify.

Thank you for pointing it out, we have modified it to global warming.

14. Line 39: Use "gases" instead of "gasses."

We have modified it, thank you for pointing it out.

15. Line 43: Provide the full names of TROPOMI, GOSAT, and SCIAMACHY before using the acronyms.

We have modified it, thank you for pointing it out.

16. Line 44: Spell out the full forms of AGAGE, WMO, and TCCON.

We have modified it, thank you for pointing it out.

17. Line 50: Use "extensive work to estimate uncertainty" instead of "extensive work on uncertainty."

We have modified it, thank you for pointing it out.

18. Line 58: Specify what kind of in-situ processing and other processes are referred to; please explain in detail.

Here, it refers to the space-time matching of daily variation of methane in the same grid, corresponding to the time gradient term in our equation.

19. Line 62: Clarify what "significant non-linearity" means and provide quantitative uncertainty values for emission estimates.

The specific meaning of nonlinearity has been explained in Question 4 of the above specific comment.

20. Line 69: Define "clean area." Do you mean a background area without emissions?

A clean area means that there are no large emission sources such as coal mines or oil and gas fields here; it is merely a town or a village.

21. Line 72: Specify the type of threshold and filter used, or refer to the section where this is detailed.

Thank you for your point. We have already explained it in Section 2.4

22. Lines 87–88: Clarify the phrase "The XCH4 retrieval used herein (version 2.4.0 Level 2) relies on a physical algorithm that factors in surface and atmospheric scattering." What exactly does this mean?

Here it means the methane column concentration product inverted by TROPOMI using a fully physical algorithm.

Reference

Cohen, J. B and Wang, C.: Estimating global black carbon emissions using a top-down Kalman Filter approach, JGR Atmospheres, 119(1), https://doi.org/10.1002/2013JD019912, 2014.

Dubey, L., Cooper, J., Hawkes, A.: Minimum detection limits of the TROPOMI satellite sensor across North America and their implications for measuring oil and gas methane emissions, Science of The Total Environment, 872 (10), https://doi.org/10.1016/j.scitotenv.2023.162222, 2023.

He, Q., Qin, K., Li, X., Lu, L., Wong, M. S., & Cohen, J. B. (2025). Diurnal NOx emission underestimation constrained using overlapping TROPOMI swaths. Atmospheric Environment, 358, 121354. https://doi.org/10.1016/j.atmosenv.2025.121354

Hemati, M., Mahdianpari, M., Nassar, R. et al. Urban methane emission monitoring across North America using TROPOMI data: an analytical inversion approach. Sci Rep 14, 9041 (2024). https://doi.org/10.1038/s41598-024-58995-8.

Hu, W., Qin, K., Lu, F., Li, D., and Cohen, J. B.: Merging TROPOMI and eddy covariance observations to quantify 5-years of daily CH4 emissions over coal-mine dominated region, Int J Coal Sci Technol, 11, 56, https://doi.org/10.1007/s40789-024-00700-1, 2024.

Jacob, D. J., Turner, A. J., Maasakkers, J. D., Sheng, J., Sun, K., Liu, X., Chance, K., Aben, I., McKeever, J., and Frankenberg, C.: Satellite observations of atmospheric methane and their value for quantifying methane emissions, Atmos. Chem. Phys., 16, 14371–14396, https://doi.org/10.5194/acp-16-14371-2016, 2016.

Li, X., Cohen, J. B., Qin, K., Geng, H., Wu, X., Wu, L., Yang, C., Zhang, R., and Zhang, L.: Remotely sensed and surface measurement- derived mass-conserving inversion of daily NOx emissions and inferred combustion technologies in energy-rich northern China, Atmos. Chem. Phys, 23, 8001–8019, https://doi.org/10.5194/acp-23-8001-2023, 2023.

Liu, J., Cohen, J. B., Tiwari, P., Liu, Z., Yim, S. H. L., Gupta, P., and Qin, K.: New top-down estimation of daily mass and number column density of black carbon driven by OMI and AERONET observations, Remote Sensing of Environment, 315, 114436, https://doi.org/10.1016/j.rse.2024.114436, 2024.

Liu, M., van der A, R., van Weele, M., Eskes, H., Lu, X., Veefkind, P., de Laat, J., Kong, H., Wang, J., Sun, J., Ding, J., Zhao, Y., and Weng, H.: A New Divergence Method to Quantify Methane Emissions Using Observations of Sentinel-5P TROPOMI, Geophysical Research Letters, 48(18), e2021GL094151, https://doi.org/10.1029/2021GL094151, 2021.

Liu, Z., Cohen, J. B., Wang, S., Wang, X., Tiwari, P., and Qin, K.: Remotely sensed BC columns over rapidly changing Western China show significant decreases in mass and inconsistent changes in number, size, and mixing properties due to policy actions, npj Clim Atmos Sci, 7, 124, https://doi.org/10.1038/s41612-024-00663-9, 2024.

Lu, F., Qin, K., Cohen, J. B., He, Q., Tiwari, P., Hu, W., Ye, C., Shan, Y., Xu, Q., Wang, S., and Tu, Q.: Surface-observation-constrained high-frequency coal mine methane emissions in Shanxi, China, reveal more emissions than inventories, consistent with satellite inversion, Atmos. Chem. Phys., 25, 5837–5856, https://doi.org/10.5194/acp-25-5837-2025, 2025.

Lu, L., Cohen, J. B., Qin, K., Li, X., and He, Q.: Identifying missing sources and reducing NOx emissions uncertainty over China using daily satellite data and a mass-conserving method, Atmos. Chem. Phys., 25, 2291–2309, https://doi.org/10.5194/acp-25-2291-2025, 2025.

Qin, K., Lu, L., Liu, J., He, Q., Shi, J., Deng, W., Wang, S., and Cohen, J. B.: Model-free daily inversion of NOx emissions using TROPOMI (MCMFE-NOx) and its uncertainty: Declining regulated emissions and growth of new sources, Remote Sensing of Environment, 295, 113720, https://doi.org/10.1016/j.rse.2023.113720, 2023.

Shen, L., Jacob, D.J., Gautam, R. et al. National quantifications of methane emissions from fuel exploitation using high resolution inversions of satellite observations. Nat Commun 14, 4948 (2023). https://doi.org/10.1038/s41467-023-40671-6.

Sicsik-Paré, A., Fortems-Cheiney, A., Pison, I., Broquet, G., Opler, A., Potier, E., Martinez, A., Schneising, O., Buchwitz, M., Maasakkers, J. D., Borsdorff, T., and Berchet, A.: Can we obtain consistent estimates of the emissions in Europe from three different CH4 TROPOMI products?, EGUsphere [preprint], https://doi.org/10.5194/egusphere-2025-2622, 2025.

Tiwari, P., Cohen, J. B., Lu, L., Wang, S., Li, X., Guan, L., Liu, Z., Li, Z., and Qin, K.: Multi-platform observations and constraints reveal overlooked urban sources of black car bon in Xuzhou and Dhaka, Commun. Earth. Environ, 6, 38, https://doi.org/10.1038/s43247-025-02012-x, 2025.

Tiwari, P., Cohen, J.B., Wang, X., Wang, S., and Qin, K.: Radiative forcing bias calculation based on COSMO (Core-Shell Mie model Optimization) and AERONET data, npj Clim Atmos Sci, 6, 193, https://doi.org/10.1038/s41612-023-00520-1, 2023.

Torres, O., Jethva, H., Ahn, C., Jaross, G., and Loyola, D. G.: TROPOMI aerosol products: evaluation and observations of synoptic-scale carbonaceous aerosol plumes during 2018–2020, Atmos. Meas. Tech, 13, 6789–6806, https://doi.org/10.5194/amt-13-6789-2020, 2020.

Varon, D. J., Jacob, D. J. et al., Seasonality and declining intensity of methane emissions from the Permian and nearby US oil and gas basins. [Preprint.] https://doi.org/10.31223/X56B2G, 2025.

Varon, D. J., Jacob, D. J., Hmiel, B., Gautam, R., Lyon, D. R., Omara, M., Sulprizio, M., Shen, L., Pendergrass, D., Nesser, H., Qu, Z., Barkley, Z. R., Miles, N. L., Richardson, S. J., Davis, K. J., Pandey, S., Lu, X., Lorente, A., Borsdorff, T., Maasakkers, J. D., and Aben, I.: Continuous weekly monitoring of methane emissions from the Permian Basin by inversion of TROPOMI satellite observations, Atmos. Chem. Phys., 23, 7503–7520, https://doi.org/10.5194/acp-23-7503-2023, 2023.

Wang, S., Wang, X., Cohen, J. B and Qin, K.: Inferring Polluted Asian Absorbing Aerosol Properties Using Decadal Scale AERONET Measurements and a MIE Model, Geophysical Research Letters, 48(20), https://doi.org/10.1029/2021GL094300, 2021.

---

## Author Response (AR2)

**Dear Editors and Reviewers**

The point of this work is based on the fact that all observations have uncertainty, and due to the non-linear nature of atmospheric mass conservation, the computations which convert observed atmospheric column concentrations and associated uncertainty into emissions produce a non-linear response on computed emission uncertainty. We specifically aim to conservatively yet explicitly include these uncertainties on the emissions computed, so as to yield a realistic and reasonable final product, with a reasonable range of values.

To our knowledge, this is the first work done which has attempted to apply this basic science-philosophy approach to estimate emissions of CH4 using observations of XCH4 from satellite. Our group has just published a paper using a similar approach (with quite different chemical and dynamical driving terms) to estimate emissions of NOx using observations of column NO2 also from TROPOMI (Lu et al., 2025a). In general, we know that the uncertainties in the retrieval of the XCH4 will be done to uncertainties and incomplete scientific knowledge of clouds, aerosol, and surface terms. We know that these uncertainties must be smaller than the uncertainties of retrieved gas products, since the total impact on radiation streams from these three is an order of magnitude larger than that of the gasses.

We have specifically published work using aerosol products from TROPOMI, OMI, and MISR where we demonstrate that the aerosol uncertainty is at least 5% (Tiwari et al., 2025; Liu J et al., 2024; Liu Z et al., 2024) and demonstrate using column and surface observations that the variability on inverted radiative products is also at least of a similar amount (Tiwari et al., 2023; Guan et al., 2025). In general, we have found that TROPOMI NO2 products, which are considered one of the most reliable gaseous products, has an uncertainty ranging from at least 10% (Qin et al., 2023; Lu et al., 2025b; Wang et al., 2025; Li et al., 2023), to possibly as high as 40% or more (Boersma et al., 2018; Pollard et al., 2022).

Additionally, the differences between the observations from TROPOMI and from the long-term WMO surface station are up to nearly 10% as explained in the initial version of the paper published. We now also include air-core observations and find that these closely match the surface observations and also have a difference of around 10% (or possibly even more) when compared with TROPOMI. Given all of the above reasons, we believe that using a 10% uncertainty for XCH4 is reasonable.

We specifically believe that we should not need to compute the exact uncertainty for each contributing factor in the inversions, as this is an issue with the underlying physics based radiative transfer model itself. It is interesting work, and we hope that those who produce the column products can continue to pursue ever improved concentration

inversions, which better represent the underlying uncertainties. The point is that uncertainty is meant to account for both what is known, as well as other sources of uncertainty in terms of what is unknown, or what is currently not able to be measured.

In response to the reviewer's comments, we have provided comparisons with aerosol extinction, aerosol absorption, and surface albedo, and demonstrate that in fact that our emissions are found to be robust and reliable in areas where these factors all have less of an impact on the retrieval, while many of the grids found to not have reliable emissions have a larger impact from these driving factors. This demonstrates that our philosophical approach is in fact reasonable. The community still does not fully understand the physics and composition of the atmosphere, and therefore there are still physical mechanisms and calculation approaches, as well as parts of the atmospheric composition which in reality are related to the retrieval of XCH4 but are still not certain. This applies not only to the aerosol extinction, aerosol absorption, and surface albedo, but also to other factors we currently cannot readily observe including water vapor, and carbon monoxide, which are currently not well known.

For these reasons, we believe that our approach is reasonable and conforms to high standards of scientific enquiry. We do believe in satellite products and want the community to feel confident in their use. For this reason, our work proposes an unbiased framework by which a reasonable accounting of their uncertainty can be robustly applied, and therefore the community can gain more confidence in satellite observation use to solve real-world problems.

**Author response – RC3**
We thank the reviewer for the thoughtful comments provided. We respond to them below in the following manner: the comments are directly copied in black, our author responses in blue, and suggested new manuscript text is indicated in green. New line numbers in the revised manuscript are provided

**1.Regarding QA=1.0**
• **If the area does not have enough observations after applying QA = 1.0, it clearly indicates that the data are not sufficient for this test. I had suggested adding additional filters such as AOD and albedo, together with QA > 0.5, to help avoid possible artefacts. However, the author treated this approach as merely a philosophical choice to avoid artefacts, which is not correct.**

Our paper is attempting to quantify a way by which an objective analysis can be performed on retrieved XCH4 values and their uncertainty in tandem, so as to invert CH4 emissions with sufficient precision, that the inverted emissions are not merely an

artefact of the observed XCH4 noise. We fully agree that pixels with unusual albedo and aerosol interactions used for the inversion of XCH4 concentration are pixels which likely will have more uncertainty. What we have done is to use all of the data which has a QA>0.5, since this is what the community currently does as a standard. This was already mentioned in the previous version of the reply comments, please refer to the first response of comment to reviewer 3.

We specifically argue that AOD is insufficient, and claim that knowledge of the AOD at a minimum of two wavebands, merged with information of either SSA or AAOD at a minimum of two wavebands is in fact required to determine whether or not aerosols are substantially contributing to issues with XCH4 retrieval (i.e., Tiwari et al., 2023; Liu et al., 2024; Tiwari et al., 2025; Liu, J et al., 2024; Guan et al., 2025). For this reason, the version of the paper you reviewed in Figure S1 already included information about aerosols. In specific we demonstrated that the AAOD at 440nm, the AAOD at 880nm, and the ratio of the two AAODs (as observed from MISR over the region) following (Liu et al., 2025) can provide insight. Areas with a large AAOD ratio indicate there is more larger absorbing aerosol, which is more likely to impact the radiation wavebands which are used in the inversion of XCH4.

To further address your concerns, we have done additional analysis. We observe that the values of the ratio are somewhat larger over areas where we have filtered the emissions as being non-realistic [filtered points] (such as area A in Figure 4a). Furthermore, we observe that most of the grids where we have valid emissions [retained points] are located are in areas with very low AAOD ratios, as demonstrated in the new Figure S1g below. Therefore, our approach is successful in determining clearly that the few pixels impacted by aerosols in the region are removed using this process.

To further demonstrate the effectiveness of our method, we provide the PDF of the ratio of AAOD corresponding to the filtered points (Figure 5a), as well as the PDF of the ratio of AAOD corresponding to the retained points (Figure 5d). It can be clearly seen that the filtered emissions include some points with extremely high AAOD ratios, and most of the points with relatively high ratios of AAOD have been removed.

This result offers a physical explanation of why there is a substantial uncertainty in the XCH4 retrieved values, and strengthens our argument that 10% uncertainties are reasonable. What is also important is that our approach is capable of detecting such a non-linear uncertainty propagation, while standard emissions estimation approaches (Schneising et al., 2020; Veefkind et al., 2023; Hancock et al., 2025) in fact compute negative emissions, without realizing that the emissions are merely due to observational uncertainty.

[Figure]

Figure S1: (a) Annual average methane column concentration of Waliguan region; (b)TROPOMI XCH4 observational density in Waliguan region; (c) Surface pressure in the Waliguan region; (d) The average surface albedo of Waliguan region; (e) The average AAOD_440nm of Waliguan region; (f) The average AAOD_880nm of Waliguan region; (g) The ratio of AAOD_440nm to AAOD_880nm; (h) The average AOD_865nm of Waliguan region. The black box represents the grid where our retained valid emissions are located

[Figure]

Figure 5: (a), (b), and (c) are the PDFs of the ratio of MISR AAOD observed at 443nm to AAOD observed at 865nm, MISR AOD observed at 865nm, and albedo corresponding to the location of the filtered emission (invalid emissions), respectively; while (d), (e), and (f) are the same respective values, but corresponding to the location of the retained emissions (valid emissions), respectively.

The above content has been supplemented in the manuscript, corresponding to lines 288 to 294: "Next, we examine the impact of aerosol absorption based on MISR

AAOD at 443nm, 865nm, and the ratio of the two AAODs as given in (Figures S1e, f, g) following Liu, Z et al. (2025). Areas with a large AAOD ratio indicate the presence of larger-sized absorbing aerosol, which is more likely to impact the radiation wavebands used in the inversion of XCH4. We observe that the ratio is somewhat larger over areas with emissions we have filtered (such as area A in Figure 4a). Furthermore, we observe that most of the grids with valid emissions are located are in areas with very low AAOD ratios. The PDFs of the AAOD ratio for the invalid and valid emission points, shown in Figure 5a and d respectively. Therefore, our approach is successful in determining that the pixels more impacted by aerosols are in fact filtered."

**Figure 4(d) and 4(e) show high AOD in this area, which likely introduced bias in the data and resulted in negative emissions.**

We would like to clarify that the reviewer's comment is based on a misunderstanding. As described above, the presented plots are of AAOD, not AOD. AOD reflects the extinction, which changes the entirety of the stream of energy (Liu et al., 2024; Tiwari et al., 2023, 2025), while AAOD reflects the absorption, which affects the line-by-line radiance absorption of the XCH4 when inverted using Beer's Law, DOAS, or similar physics-based techniques (Kuhlmann et al., 2025; Tian et al., 2021).

We have added in specific data of the AOD (from MISR at 865nm). To display the areas with higher AOD more clearly, we have removed the values with AOD less than 0.3, as shown in Figure S1h, the AOD of the locations where our valid emissions are located is almost all less than 0.3. Figures 5b and e show the PDFs of AOD corresponding to the points of the invalid and valid emissions, respectively. Although the AOD in both Figure 5b and e is relatively small (all less than 0.11), the valid emissions have less AOD (MISR) than invalid emissions.

We also added TROPOMI AOD_SWIR as shown in Figure S2, The AOD at the location where our valid emissions are located remains very low, and the valid emissions have less medium level of AOD than invalid emissions as show in Figure S2a and b.

[Figure]

Figure S2: (a), (b), are the PDFs of the TROPOMI AOD_SWIR corresponding to the points of the invalid emission and valid emissions, respectively; (c) The average TROPOMI AOD_SWIR of Waliguan region. The black box represents the grid where our retained valid emissions are located

The above content has been supplemented in the manuscript, corresponding to lines 277 to 287: "Aerosols impact XCH4 in two different ways, through scattering and through absorption. First, aerosols increase radiative scattering, changing the entirety of the stream of energy (Kahn et al., 2023; Liu et al., 2024; Tiwari et al., 2023, 2025), while also absorbing radiation, affecting line-by-line radiance absorption used to invert XCH4 based on Beer's Law, DOAS, or similar physics-based techniques (Kuhlmann et al., 2025; Tian et al., 2021; Guan et al., 2025). We have included observations of AOD at a band as close to that retrieved from TROPOMI as possible (specifically observed by MISR at 865nm). As shown in Figure S1h, almost all of our valid emissions occur at locations where the 2019-2021 average AOD is less than 0.3. Figures 5b and e show the PDFs of AOD corresponding to the points in space and time of the invalid and valid emissions, respectively, while spatial plots (see Figure S2b and e) detail that while all grids are low, that those grids with valid emissions have lower AOD than grids with invalid emissions. Furthermore, we have analyzed the TROPOMI AOD_SWIR product (Figure S2), and found similarly that where our emissions are valid, the AOD remains both very low, as well as being smaller than the AOD on the invalid emissions locations, as show in Figures S2a and b."

These figures have been added to the manuscript and supplementary file, corresponding to Figure S5, Figure S1, and Figure S2.

**How can we trust these values? Additionally, the author mentions that this is a desert area. I requested albedo images, but they were not included in the revised version.**

We have included the albedo values. As shown in Figure S1d, our retained emissions do not occur at locations with either very low or very high surface albedo. Moreover, compared with the retained emissions, the surface albedo corresponding to the points where the filtered emissions are located are typically at the extreme ends of the albedo range. These invalid emissions have been effectively filtered out by our method.

The above content has been supplemented in the manuscript, corresponding to lines 295 to 297: "The surface albedo in this region as shown in Figure S1d. Our retained emissions do not occur at locations with either very low or very high surface albedo. Moreover, compared with the retained emissions (Figures 5c and f), the surface albedo corresponding to the filtered emissions are typically found closer to the extreme ends of the albedo range."

**In coastal, the methane remote sensing community has recognized both low and high albedo values when using QA > 0.5. How are such artefacts assigned in this study?**

In this work, we did not have emissions on the water region. We only have one emission grid on the coast, and the ground albedo on this grid is substantially larger than over the water region.

However, these factors alone cannot explain the total uncertainty, which may be influenced by other parameters, such as cloud, sensor and waveband resolution issues, etc. We assign reasonable uncertainty and use all data QA > 0.5, which we believe represents the most appropriate balance between data quality and coverage, and will support more future physically based research.

The above content has been supplemented in the manuscript, corresponding to lines 268 to 276: "The point of the filtering is to determine whether or not observational uncertainty or noise was the source of the retrieved emissions, or if the retrieved emissions were due to a physical signal. Some physical factors which contribute to signal observational uncertainty or noise which still exist in the QA>0.5 data include but do not fully filter for thin clouds and aerosol layers as well as moderate variations in water vapor, or medium-low albedo (Hu et al., 2016). This QA level also does not consider aerosol absorption or apply any constraints on co-absorbing gasses including but not limited to carbon monoxide. However, this work specifically analyzed a some of these drivers, and clearly determined that some of these drivers in fact contributed to those grids which were filtered (i.e., areas in which the emission derived from the signal

were negative or sufficiently small or large and positive to be considered noise)."

Lines 298 to 308: "These factors alone cannot explain the total uncertainty, which may be influenced by other parameters, such as cloud, sensor and waveband resolution issues, etc. This result offers a physical explanation of why there is a substantial uncertainty in the XCH4 retrieved values. What is also important is that our approach is capable of detecting such a non-linear uncertainty propagation, while standard emissions estimation approaches (Schneising et al., 2020; Veefkind et al., 2023; Hancock et al., 2025) in fact compute negative emissions, without realizing that the emissions are merely due to observational uncertainty. To be clear, this approach herein is further validated, since applying the uncertainty in general does actually capture a subset of physical driving factors which are expected to lead to greater retrieval noise."

**2.Ground-based measurement comparison with TROPOMI total columns**
**• Although the station is located at high altitude and represents free-tropospheric concentrations (considered as WMP background concentration), surface measurements cannot be directly compared with total column measurements, nor should their differences be treated as errors in the TROPOMI concentration. If the goal is to compare a ground-based station with total column measurements, air core measurements should be performed, as discussed in this paper: https://amt.copernicus.org/articles/10/2163/2017/**

We have compared the surface observations with air core observations as given in Figure 2 of Tao et al., (2024), which actually overlap on a day with TROPOMI and surface data, and are located close to our region of interest.

Tao et al (2024) used AirCore-measured CH4 profile compare to the spatially and temporally nearest satellite-measured (L2) and simulated CH4 profiles and found that the satellite significantly underestimated the CH4 enhancement. This difference is consistent with what is shown in our Figure 1.

We have found that the air core vertical observations and uncertainty bound match very well with the surface observations that we are using. This again supports that based on measurements, a 10% uncertainty applied to TROPOMI over the region of interest is reasonable.

The above content has been supplemented in the manuscript, corresponding to lines 217 to 220: "Observed vertical CH4 profiles were made by Tao et al. (2024) Figure 2 using AirCore at the same time and very close to where this work's satellite observations were analyzed. The results compare closely with the surface observations

made at the WMO station in Waliguan, validating that the surface observations in this region are a reasonable representation of the column average values, as shown in Figure 1."

**3. Method application and uncertainty quantification**
**• The method applied in this study requires proper uncertainty quantification, which is currently lacking. Important aspects such as wind fields,**

We believe that this statement is consistent with the methodology as explained on line 104 where we have used multiple different wind fields, possibly being mis-understood. We used daily wind data across the range from 550 to 750hPa. Our approach is a model-free analysis which actively calculates first order wind-based advection and pressure-based advection, as well as first order diffusion, similar to WRF, GEOS, and other regional and global scale models. We likely believe that the reviewer may not have realized that we do not use wind linearly, as we have stated in Eq (2), although many satellite based approximation of emissions in fact do just apply wind as a linear correction factor. Our calculation is based on the gradient of XCH4 as well as the gradient of the wind, which means that the propagation of white noise is due to the nonlinear propagation associated with the gradient term, not the linear wind field, as described in Eq (2). Due to the significant terrain changes in this area, we select the wind at the corresponding altitude level based on the altitude of each grid. However, we have already demonstrated in previous work for $NO_x$, CO, BC and CH4 emissions in polluted regions (Qin et al., 2023; Lu et al., 2025; Li et al., 2023; Li et al., 2025; Hu et al., 2024) that while this issue is relevant, that it is less impactful than using the gradient term correctly, and in the case of $NO_x$, the issue of retrieval error. This paper is the first attempt to bring the issue of retrieval error into the retrieval of CH4 emissions.

**the choice of mean values instead of medians,**

In the previous reply, we have explained the use of the median to replace the spatial filtering in the first step. Please refer to the fifth response to reviewer 3 in the previous version of the reply comments. In fact, we have always retained all of the data and can produce many different statistics, please see https://figshare.com/s/058f7f73953264e0d439.

**and the impact of different filtering procedures should be included to ensure robustness and reliability of the results.**

We have displayed the effects of the different filtering procedures clearly in Figures 3. This includes grid boxes versus none, filtering steps, etc. As shown in Figure S1,

Through the spatial filtering in the first step, we have filtered out invalid emissions from areas that are significantly affected by aerosols, albedo, and other factors. In the previous version, we have conducted a sensitivity analysis on the selection of the threshold for the first step of spatial filtering. After the first step of filtering, based on the theory of nonlinear propagation of satellite observation uncertainty in the gradient term, we carry out the second step of temporal filtering. We have already provided a detailed explanation in our previous response on how uncertainty is transferred to the emission results through the gradient term (refer to the fourth comment to reviewer 3 in the previous version of the reply comments).

**4. Comparison with existing estimates**
**• I strongly suggest comparing the results with bottom-up emission estimates and inversion-based estimates over the Permian Basin. This would provide valuable context and help validate the findings.**

In the previous version of the response, we have already provided the bottom-up emission inventory of the Waliguan region and found that there were sources missing in the inventory (EDGAR). The bottom-up emission inventory (EDGAR) of the Permian Basin is shown in Figure S6a, and Figure S6b shows the difference between our emission results and EDGAR, with the difference ranging from -285 to 3830 kg/h/grid, and the 95th percentiles of the differences is 2920 kg/h/grid. The EDGAR emission inventory is significantly lower than our results, in part due to many emissions grids missing from their dataset. We also compared the differences between the inventory of Cusworth et al. and EDGAR, as shown in Figure S6c. The difference range is from -3240 to 8440 kg/h/grid, and the 95th percentiles of the differences is 2450 kg/h/grid, which is close to the difference range between our results and EDGAR.

[Figure]

Figure S6: (a) The average annual emission flux of all sectors of EDGAR; (b) The difference between our retained valid emissions and EDGAR in 2019; (c) The difference between Cusworth et al., and EDGAR in 2019.

The above content has been supplemented in the manuscript, corresponding to lines 410 to 418: "To better compare with other inventories many of which use emissions in terms of mass per time per grid, we first convert emissions fluxes (μg/m²/s) into

emission rates over each entire TROPOMI grid (kg/h/grid). The EDGAR emission inventory over the Permian Basin is shown in Figure S6a, and Figure S6b shows the difference between our emission results and EDGAR, with the grid-by-grid ranging from -285 to 3830 kg/h/grid, and the 95th percentiles of the difference is 2920 kg/h/grid. The EDGAR emission inventory is significantly lower than our results, in part due to many emissions grids missing from their dataset, although their grid with the highest emission is still lower than our result, indicating that our approach does not have a high bias. We also compared the differences between the inventory of Cusworth et al. (2021) and EDGAR, as shown in Figure S6c. The difference range is from -3240 to 8440 kg/h/grid, and the 95th percentiles of the difference is 2450 kg/h/grid, which is close to the difference range between our results and EDGAR." The figure has been added to the supplementary file, corresponding to Figure S6

**5. Clarification on comparison with Cusworth et al. (2021)**

**• In Line 575, the manuscript states:**

**"It's clear that our results are larger, which is consistent with the observations made by aircraft having scan widths of 3 and 4.5 km, which is always smaller than our grid resolution of 0.05° × 0.05° (~5 km). This means that even if their scan crossed the center of our grid, our grid would still contain information outside of their scan width, and if the scan only crossed a small amount of our grid, then the grid would contain far more information."**

**Instead of only stating that the results are consistent, please quantify the differences between your results and those of Cusworth et al. (2021). For example, provide a plot of the standard deviation or relative differences to show how your estimates compare with theirs. This would strengthen the credibility of the comparison.**

Thank you very much for your suggestion. The construction of this model is based on the mass conservation equation. By fitting the ground observation data as physical constraints with the TROPOMI XCH4 data, a total of 900 sets of fitting coefficients were obtained. Note that these fitting coefficients are physical variables, specifically representing the time and weight scales of diffusion and advection. Based on these coefficients, 900 emission values can be estimated for each grid point. We take the average of the 450 values within the 20th to 80th percentile range as the final emission estimate for this point, and the uncertainty range is defined as the 20th and 80th percentiles of the emission value sequence. This range is meant to widely sample more standard observed conditions. Unfortunately, Cusworth did not provide high spatial and temporal resolution wind, pressure, and concentration data, or we could choose this range with more certainty (i.e. Lu, F et al., 2025).

According to the above logic, we have redrawn the scatter plots of our results and the observed emissions of Cusworth et al. The red dots in the scatter plot of Figure 8b represent the points where our emission results overlap with Cusworth (2021) in both time and space, with the R is 0.8, indicating a good match, and the MAE is 660 kg/h/grid. The black dots and blue dots respectively represent our effective emissions and the emission estimates of Cusworth et al. at other times in the grid where the red dots are located. We found that the methane emissions from these emission facilities varied significantly over time.

[Figure]

Figure 8: (a) Annual mean CH4 emissions (kg/h/grid) on grids which overlap spatially with an emission observed by observed by Cusworth et al. (2021); (b) The red dots in the scatter plot represent the points where our retained valid emissions overlap with those of Cusworth et al. (2021) in both space and time and the difference is within 2 times. The black and blue dots denote emission estimates from our study and Cusworth et al. (2021) respectively, which share spatial overlap but differ temporally. The gray error bars represent the associated uncertainties. Backgrounds of (a) is from Esri World Imagery.

[revised manuscript text omitted]

Lu et al., "New Perspective on Using Observational Uncertainty to Improve Reliability of NOx Emissions Over Northern China," in IEEE Transactions on Geoscience and Remote Sensing, https://doi.org/10.1109/TGRS.2025.3620116,2025b.

Lu, F., Qin, K., Cohen, J. B., He, Q., Tiwari, P., Hu, W., Ye, C., Shan, Y., Xu, Q., Wang, S., and Tu, Q.: Surface-observation-constrained high-frequency coal mine methane emissions in Shanxi, China, reveal more emissions than inventories, consistent with satellite inversion, Atmos. Chem. Phys., 25, 5837–5856, https://doi.org/10.5194/acp-25-5837-2025, 2025.

Lu, L., Cohen, J. B., Qin, K., Li, X., and He, Q.: Identifying missing sources and reducing NOx emissions uncertainty over China using daily satellite data and a mass-conserving method, Atmos. Chem. Phys., 25, 2291–2309, https://doi.org/10.5194/acp-25-2291-2025, 2025a.

Pollard, D. F., Hase, F., Sha, M. K., Dubravica, D., Alberti, C., and Smale, D.: Retrievals of XCO2, XCH4 and XCO from portable, near-infrared Fourier transform spectrometer solar observations in Antarctica, Earth Syst. Sci. Data, 14, 5427–5437, https://doi.org/10.5194/essd-14-5427-2022, 2022.

Qin, K., Lu, L., Liu, J., He, Q., Shi, J., Deng, W., Wang, S., and Cohen, J. B.: Model-free daily inversion of NOx emissions using TROPOMI (MCMFE-NOx) and its uncertainty: Declining regulated emissions and growth of new sources, Remote Sensing of Environment, 295, 113720, https://doi.org/10.1016/j.rse.2023.113720, 2023.

Schneising, O., Buchwitz, M., Reuter, M., Vanselow, S., Bovensmann, H., and Burrows, J. P.: Remote sensing of methane leakage from natural gas and petroleum systems revisited, Atmos. Chem. Phys., 20, 9169–9182, https://doi.org/10.5194/acp-20-9169-2020, 2020.

Tao, M., Cai, Z., Zhu, S. et al.: New evidence for CH4 enhancement in the upper troposphere associated with the Asian summer monsoon. Environmental Research Letters, 19, 034033, https://doi.org/10.1088/1748-9326/ad2738, 2024.

Tian, X., Wang, Y., Beirle, S., Xie, P., Wagner, T., Xu, J., Li, A., Dörner, S., Ren, B., and Li, X.: Technical note: Evaluation of profile retrievals of aerosols and trace gases for MAX-DOAS measurements under different aerosol scenarios based on radiative transfer simulations, Atmos. Chem. Phys., 21, 12867–12894, https://doi.org/10.5194/acp-21-12867-2021, 2021.

Tiwari, P., Cohen, J. B., Lu, L., Wang, S., Li, X., Guan, L., Liu, Z., Li, Z., and Qin, K.: Multi-platform observations and constraints reveal overlooked urban sources of black car bon in Xuzhou and Dhaka. Commun. Earth. Environ, 6, 38, https://doi.org/10.1038/s43247-025-02012-x, 2025.

Tiwari, P.; Cohen, J.B.; Wang, X.; Wang, S.; Qin, K. Radiative forcing bias calculation based on COSMO (Core-Shell Mie model Optimization) and AERONET data, npj Clim Atmos Sci, 6, 193. https://doi.org/10.1038/s41612-023-00520-1, 2023.

Veefkind, J. P., Serrano-Calvo, R., de Gouw, J., Dix, B., Schneising, O., Buchwitz, M., Barré,J., van der A, R. J., Liu, M., and Levelt, P. F.: Widespread frequent methane emissions from the oil and gas industry in the Permian basin, Journal of Geophysical Research: Atmospheres, 128, e2022JD037479, https://doi.org/10.1029/2022JD037479, 2023.

Wang, S., Cohen, J.B., Guan, L. et al.: Observationally constrained global NOx and CO emissions variability reveals sources which contribute significantly to CO2 emissions. npj Clim Atmos Sci 8, 87, https://doi.org/10.1038/s41612-025-00977-2, 2025.